# Differential contribution of two organelles of endosymbiotic origin to iron-sulfur cluster synthesis and overall fitness in Toxoplasma

Sarah Pamukcu[1], Aude Cerutti [1], Yann Bordat[1], Sonia Hem[2], Valérie Rofidal [2], Sébastien Besteiro [3]*

1 LPHI, Univ Montpellier, CNRS, Montpellier, France, 2 BPMP, Univ Montpellier, CNRS, INRAE, Institut Agro, Montpellier, France, 3 LPHI, Univ Montpellier, CNRS, INSERM, Montpellier, France

* sebastien.besteiro@inserm.fr

**Data Availability Statement:** All raw MS data and MaxQuant files generated have been deposited to the ProteomeXchange Consortium via the PRIDE

## Abstract

Iron-sulfur (Fe-S) clusters are one of the most ancient and ubiquitous prosthetic groups, and they are required by a variety of proteins involved in important metabolic processes. Apicomplexan parasites have inherited different plastidic and mitochondrial Fe-S clusters biosynthesis pathways through endosymbiosis. We have investigated the relative contributions of these pathways to the fitness of *Toxoplasma gondii*, an apicomplexan parasite causing disease in humans, by generating specific mutants. Phenotypic analysis and quantitative proteomics allowed us to highlight notable differences in these mutants. Both Fe-S cluster synthesis pathways are necessary for optimal parasite growth in vitro, but their disruption leads to markedly different fates: impairment of the plastidic pathway leads to a loss of the organelle and to parasite death, while disruption of the mitochondrial pathway trigger differentiation into a stress resistance stage. This highlights that otherwise similar biochemical pathways hosted by different sub-cellular compartments can have very different contributions to the biology of the parasites, which is something to consider when exploring novel strategies for therapeutic intervention.

## Author summary

*Toxoplasma gondii* is a ubiquitous unicellular parasite that harbours two organelles of endosymbiotic origin: the mitochondrion, and a relict plastid named the apicoplast. Each one of these organelles contains its own machinery for synthesizing iron-sulfur clusters, which are important protein co-factors. In this study, we show that interfering with either the mitochondrial or the plastidic iron-sulfur cluster synthesizing machinery has a profound impact on parasite growth. However, while disrupting the plastidic pathway led to an irreversible loss of the organelle and subsequent death of the parasite, disrupting the mitochondrial pathway led to conversion of the parasites into a stress resistance form. We used a comparative quantitative proteomic analysis of the mutants, combined with experimental validation, to provide mechanistic clues into these different phenotypic outcomes. Although the consequences of disrupting each pathway were manifold, our data

partner repository (https://www.ebi.ac.uk/pride/) with the dataset identifier PXD023854.

**Funding:** AC was supported by a fellowship from the Fondation pour la Recherche Médicale (Equipe FRM EQ20170336725), https://www.frm.org/; SB acknowlegdes support from the Labex Parafrap (ANR-11-LABX-0024), https://www.labex-parafrap.fr/en/, and the Agence Nationale de la Recherche (ANR-19-CE15-0023), https://anr.fr/en/. The funders had no role in study design, data collection and analysis, decision to publish, or preparation of the manuscript.

**Competing interests:** The authors have declared that no competing interests exist.

highlighted potential changes at the metabolic level. For instance, the plastidic iron-sulfur cluster synthesis pathway may be important for maintaining the lipid homeostasis of the parasites, while the mitochondrial pathway is likely crucial for maintaining their respiratory capacity. Interestingly, we have discovered that other mutants severely impacted for mitochondrial function, in particular the respiratory chain, are able to survive and initiate conversion to the stress resistance form. This illustrates a different capacity for *T. gondii* to adapt for survival in response to distinct metabolic dysregulations.

## Introduction

Endosymbiotic events were crucial in the evolutionary timeline of eukaryotic cells. Mitochondria and plastids evolved from free-living prokaryotes that were taken up by early eukaryotic ancestors and transformed into permanent subcellular compartments that have become essential for harnessing energy or synthesizing essential metabolites in present-day eukaryotes [1]. As semiautonomous organelles, they contain a small genome, but during the course of evolution a considerable part of their genes have been transferred to the cell nucleus. Yet, they rely largely on nuclear factors for their maintenance and expression. Both organelles are involved in critically important biochemical processes. Mitochondria, which are found in most eukaryotic organisms, are mostly known as the powerhouses of the cell, owing to their ability to produce ATP through respiration. Importantly, they are also involved in several other metabolic pathways [2], including the synthesis of heme groups, steroids, amino acids, and iron-sulfur (Fe-S) clusters. Moreover, they have important cellular functions in regulating redox and calcium homeostasis. Similarly, plastids that are found in plants, algae and some other eukaryotic organisms, host a diverse array of pathways that contribute greatly to the cellular metabolism [3]. While often identified mainly as compartments where photosynthesis occurs, plastids host other important metabolic pathways. For example, they are involved in the assimilation of nitrogen and sulfur, as well as the synthesis of carbohydrates, amino acids, fatty acids and specific lipids, hormone precursors, and also Fe-S clusters. The best-characterized plastid is arguably the plant cell chloroplast, but not all plastids have photosynthetic function, and in land plants they are in fact a diverse group of organelles that share basal metabolic pathways, but also have specific physiological roles [4].

The phylum Apicomplexa comprises a large number of single-celled protozoan parasites responsible for serious disease in animals, including humans. For example, this phylum includes parasites of the genus *Plasmodium* that are responsible for the deadly malaria, and *Toxoplasma gondii* a ubiquitous parasite that can lead to a severe pathology in immunocompromised individuals. Apicomplexan parasites evolved from a photosynthetic ancestor and many of them still retain a plastid [5,6]. This plastid, named the apicoplast, originated from a secondary endosymbiotic event [7,8]. It has lost its photosynthetic properties as the ancestors of Apicomplexa switched to an intracellular parasitic lifestyle [9]. The apicoplast nevertheless still hosts four main metabolic pathways [10,11]: a so-called non-mevalonate pathway for the synthesis of isoprenoid precursors, a type II fatty acid synthesis pathway (FASII), part of the heme synthesis pathway, and a Fe-S cluster synthesis pathway. As the apicoplast is involved in these important biological processes for the parasite, and as they markedly differ from those of the host (because of their algal origin), that makes it a valuable potential drug target. Apicomplexan parasites also generally contain a single tubular mitochondrion, although its aspect may vary during parasite development [12,13]. The organelle is an important contributor to the parasite's metabolic needs [14]. It classically hosts tricarboxylic acid (TCA) cycle reactions,

which are the main source of electrons that feeds the mitochondrial electron transport chain (ETC) which generates a proton gradient used for ATP production. It also contains additional metabolic pathways, like a Fe-S cluster synthesis pathway and part of the heme synthesis pathway operating in collaboration with the apicoplast. The latter reflects obvious functional links between the organelles and potential metabolic interactions, which is also illustrated by their physical connection during parasite development [15,16].

Fe-S clusters are simple and ubiquitous cofactors involved in a great variety of cellular processes. As their name implies, they are composed of iron and inorganic sulfur whose chemical properties confer key structural or electron transfer features to proteins in all kingdoms of life. They are important to the activities of numerous proteins that play essential roles to sustain fundamental life processes including, in addition to electron transfer and exchange, iron storage, protein folding, oxygen/nitrogen stress sensing, and gene regulation [17]. The synthesis of Fe-S clusters and their insertion into apoproteins requires complex machineries and several distinct pathways have been identified in bacteria for synthesizing these ancient cofactors [18]. They include the ISC (iron-sulfur cluster) pathway for general Fe–S cluster assembly [19], and the SUF (sulfur formation) pathway [20] that is potentially activated in oxidative stress conditions [21]. Eukaryotes have inherited machineries for synthesizing Fe-S cluster through their endosymbionts [22]. As a result, organisms with both mitochondria and plastids, like land plants, use the ISC pathway for assembling Fe-S clusters in the mitochondria and the SUF pathway for Fe-S clusters in the plastids [23]. Additional protein components that constitute a cytosolic Fe-S cluster assembly machinery (CIA) have also been identified: this pathway is important for the generation of cytosolic, but also of nuclear Fe-S proteins, and is highly dependent on the ISC mitochondrial pathway for providing a sulfur-containing precursor [24].

Like in plants and algae, apicoplast-containing Apicomplexa seem to harbour the three (ISC, SUF and CIA) Fe-S cluster synthesis pathways. Although the CIA pathway was recently shown to be important for *Toxoplasma* fitness [25], investigations in apicomplexan parasites have been so far mostly conducted in *Plasmodium* species and they essentially focused on the apicoplast-located SUF pathway [26–31]. The SUF pathway was shown to be essential for the viability of malaria parasites during both the erythrocytic and sexual stages of development, and has thus been recognized as a putative avenue for discovering new antiparasitic drug targets (reviewed in [32]). Contrarily to the ISC pathway, which is also present in the mammalian hosts of apicomplexan parasites, the SUF pathway may indeed yield interesting specificities that may be leveraged for therapeutic intervention. However, very little is known about Fe-S clusters synthesis in other apicomplexan parasites, including *T. gondii*. For instance, out of the four known metabolic pathways hosted by the apicoplast, Fe-S cluster synthesis was the only one remaining to be functionally investigated in *T. gondii*, while the others were all shown to be essential for the tachyzoite stage of the parasite (a fast replicating developmental stage responsible for the symptoms of the disease) [33–36]. Here, we present the characterization of two *T. gondii* mutants we generated to specifically impact the plastidic and mitochondrial SUF and ISC pathways, respectively. Our goal was to assess the relative contributions of these compartmentalized pathways to parasite development and fitness.

## Results

### TgNFS2 and TgISU1 are functional homologs of components of the Fe-S cluster synthesis pathways

Fe-S cluster biosynthesis pathways in the mitochondrion and the plastid follow a similar general pattern: cysteine desulfurases (NFS1, NFS2) produce sulfur from L-cysteine, scaffold

proteins (ISU1, SufB/C/D) provide a molecular platform allowing iron and sulfur to meet and form a cluster, and finally carrier proteins (like IscA or SufA) deliver the cluster to target apo-proteins [23]. The cytosolic CIA pathway, which is responsible for the de novo formation of Fe-S clusters to be incorporated in cytosolic and nuclear proteins, is dependent on the ISC pathway, as its first step requires the import of a yet unknown sulfur-containing precursor that is translocated to the cytosol from the mitochondrion [24]. To get a general overview of the predicted components for the Fe-S cluster machinery in *T. gondii*, we conducted homology searches in the ToxoDB.org database [37], using well-characterized proteins from plants (*Arabidopsis thaliana*) belonging to the SUF, ISC and CIA pathways (S1 Table). Data from global mapping of protein subcellular location by hyperLOPIT spatial proteomics [38] was in general in good accordance with the expected localization of the homologs (with the noticeable exception of members of the NBP35/HCF101 ATP-binding proteins). Overall, our search revealed that *T. gondii* appears to have a good conservation of all the main components of the three ISC, SUF and CIA Fe-S synthesis pathways (S1 Table and Fig 1A). Additional information available on ToxoDB.org such as scores from a CRISPR/Cas9-based genome-wide screening [39], highlighted that most components of the three pathways are predicted to be important for parasite fitness. This suggests several Fe-S proteins localizing to the endosymbiotic organelles, but also the cytosol/nucleus, are essential for the optimal growth of tachyzoites. In order to verify this, our aim was to specifically interfere with the apicoplast-localized SUF pathway or the mitochondrion-localized ISC pathway in *T. gondii* tachyzoites. More precisely, we wanted to target the homologs of *A. thaliana* NFS2 and ISU1, which are both central (and presumably essential) to their respective pathways (Fig 1A): NFS2 is a cysteine desulfurase that provides the inorganic sulfur part of plastidic Fe-S clusters, while ISU1 is a scaffold protein important for cluster assembly in the mitochondrial pathway. Interfering with these key

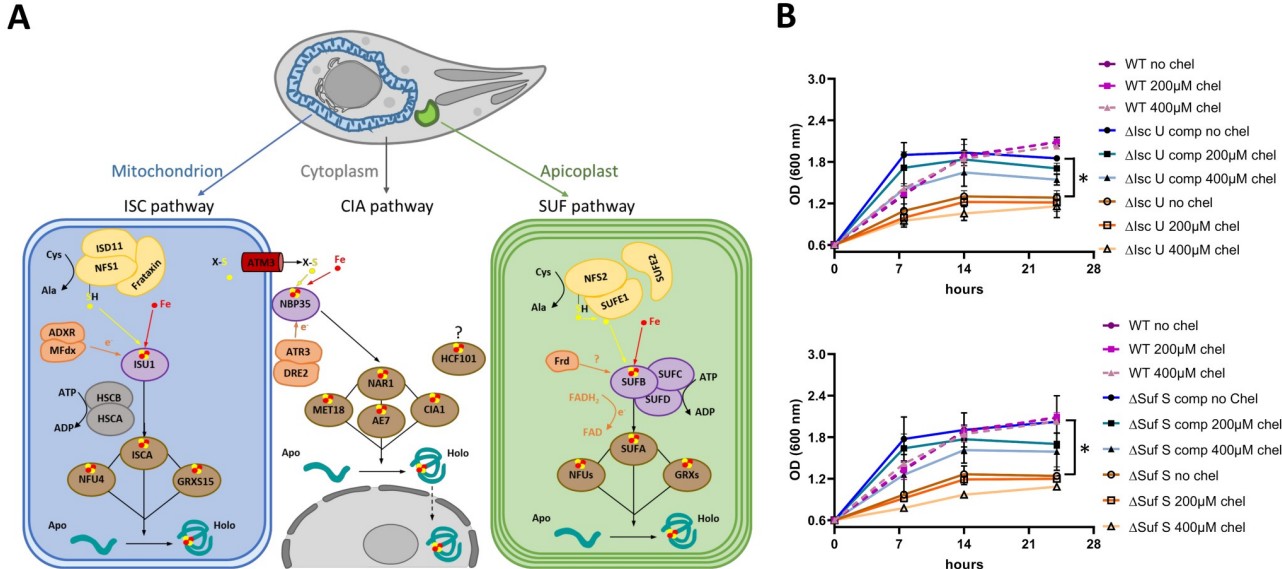

**Fig 1. TgNFS2 and TgISU1 are functional homologs of components of iron-sulfur cluster synthesis pathways.** A) Putative Fe-S cluster synthesis pathways and associated molecular machinery in *Toxoplasma*. B) Functional complementation of bacterial mutants for IscU (top) and SufS (bottom). Growth of 'wild-type' (WT) *E. coli* K12 parental strain, bacterial mutant strains and strains complemented ('comp') by their respective *T. gondii* homologues ('comp'), was assessed by monitoring the optical density at 600 nm in the presence or not of an iron chelator (2,2'-bipyridyl, 'chel'). Values are mean from $n = 3$ independent experiments ±SEM. * denotes $p \leq 0.05$, Student's *t*-test, when comparing values obtained in the absence of chelator for mutant strains versus complemented ones.

players of the upstream machinery would likely lead to a comparable disruption of the respective Fe-S cluster biogenesis pathways hosted by each organelle [40].

As a first step, we sought to determine whether TgNFS2 (TGGT1_216170) and TgISU1 (TGGT1_237560) were real functional homologs by performing complementation assays of bacterial mutants. In many bacteria, the ISC machinery is the primary system for general Fe-S cluster biosynthesis, while the SUF system plays a similar general role, but is mostly operative under stress conditions (like iron limitation or oxidative stress). In *Escherichia coli*, both pathways are partially redundant, but their individual disruption results in slowed bacterial growth, especially when limiting iron availability with a specific chelator [41]. Expression of the predicted functional domains of TgNFS2 and TgISU1 in mutant strains for the corresponding *E. coli* proteins (named SufS and IscU, respectively) improved bacterial growth, in the presence of an iron chelator or not (Fig 1B). The complementation seemed partial as complemented strains remained more sensitive to the iron chelator than the wild-type strain. Yet, and although stationary phase was reached earlier than for the wild-type bacteria, contrarily to the mutants the complemented strains showed a bacterial density close to that of the WT at stationary phase. This suggests TgNFS2 and TgISU1, in addition to a good sequence homology with their bacterial homologues (S1 Fig), have a conserved function.

We next determined the sub-cellular localizations of TgNFS2 and TgISU1 by epitope tagging of the native proteins. This was achieved in the TATi ΔKu80 cell line, which favors homologous recombination and would allow transactivation of a Tet operator-modified promoter we would later use for generating a conditional mutant in this background [42–44]. A sequence coding for a C-terminal triple hemagglutinin (HA) epitope tag was inserted at the endogenous *TgNFS2* or *TgISU1* locus by homologous recombination (S2 Fig). Using the anti-HA antibody, by immunoblot we detected two products for each protein (Fig 2A and 2B), likely corresponding to their immature and mature forms (ie after cleavage of the transit peptide upon import into the organelle). Accordingly, the analysis of TgNFS2 and TgISU1 sequences with several subcellular localization and N-terminal sorting signals site predictors confirmed they likely contained sequences for plastidic and mitochondrial targeting [45], respectively, although no consensus position of the exact cleavage sites could be determined. Immunofluorescence assay (IFA) in *T. gondii* tachyzoites confirmed HA-tagged TgNFS2 and TgISU1 co-localize with markers of the apicoplast and the mitochondrion, respectively (Fig 2C and 2D).

NFS2 is a cysteine desulfurase whose activity is enhanced by an interaction with the SUFE protein [46]. Similar to plants that express several SUFE homologues [47], there are two putative SUFE-like proteins in *T. gondii* (S1 Table), one of which was already predicted to reside in the apicoplast by hyperLOPIT (TgSUFE1, TGGT1_239320). We generated a cell line expressing an HA-tagged version of the other, TgSUFE2 (TGGT1_277010, S3A–S3C Fig), whose localization was previously unknown. Like for TgNFS2, several programs predicted a plastidic transit peptide, which was confirmed by immunoblot analysis (detecting TgSUFE2 immature and mature forms, S3D Fig). IFA showed TgSUFE2 co-localizes with an apicoplast marker (S3E Fig). This further confirms that the initial steps of Fe-S cluster biogenesis in the apicoplast are likely functionally-conserved.

## Disruption of either the plastidic or the mitochondrial Fe-S cluster pathway has a profound impact on parasite growth

In order to get insights into plastidic and mitochondrial Fe-S biogenesis, we generated conditional mutant cell lines in the TgNFS2-HA or TgISU1-HA-expressing TATi ΔKu80 background [44]. Replacement of the endogenous promoters by an inducible-Tet07*SAG4*

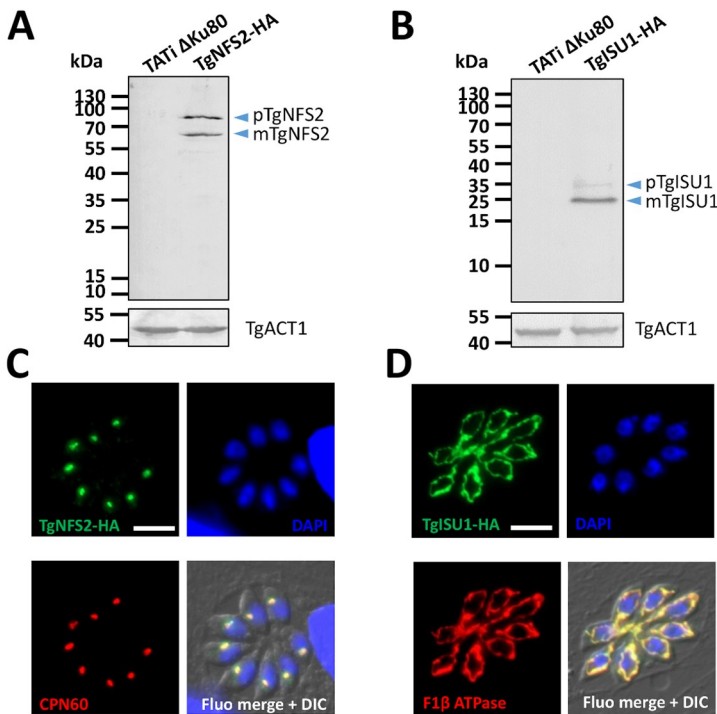

**Fig 2. TgNFS2 and TgISU1 localize to the apicoplast and the mitochondrion, respectively.** Detection by immunoblot of C-terminally HA-tagged TgNFS2 (A) and TgISU1 (B) in parasite extracts reveals the presence of both precursor (p) and mature (m) forms of the proteins. Anti-actin (TgACT1) antibody was used as a loading control. Immunofluorescence assay shows TgNFS2 co-localizes with apicoplast marker TgCPN60 (C) and TgISU1 co-localizes with mitochondrial marker F1 β ATPase (D). Scale bar represents 5 μm. DNA was labelled with DAPI. DIC: differential interference contrast.

promoter, through a single homologous recombination at the loci of interest (S4 Fig), yielded TgNFS2 and TgISU1 conditional knock-down cell lines (cKD TgNFS2-HA and cKD TgI-SU1-HA, respectively). In these cell lines, the addition of anhydrotetracycline (ATc) can repress transcription through a Tet-Off system [48]. For each cKD cell line several transgenic clones were obtained and were found to behave similarly in the initial phenotypic assays we performed, so only one was analysed further. Transgenic parasites were grown for various periods of time in presence of ATc, and protein down-regulation was evaluated. Immunoblot and IFA analyses of cKD TgNFS2 -HA and cKD TgISU1-HA parasites showed that the addition of ATc efficiently down-regulated the expression of *TgNFS2* (Fig 3A and 3C) and *TgISU1* (Fig 3B and 3D), and most of the proteins were undetectable after two days of incubation.

We also generated complemented cell lines expressing constitutively an additional copy of *TgNFS2* and *TgISU1* from the *uracil phosphoribosyltransferase* (*UPRT*) locus from a *tubulin* promoter in their respective conditional mutant backgrounds (S5A and S5B Fig). We confirmed by semi-quantitative RT-PCR (S5C Fig) that the transcription of *TgNFS2* and *TgISU1* qenes was effectively repressed in the cKD cell lines upon addition of ATc, whereas the corresponding complemented cell lines exhibited a high transcription level regardless of ATc addition (due to the expression from the strong *tubulin* promoter).

We next evaluated the consequences of TgNFS2 and TgISU1 depletion on *T. gondii* growth in vitro. First, to assess the impact on the parasite lytic cycle, the capacity of the mutants and complemented parasites to produce lysis plaques was analyzed on a host cells monolayer in

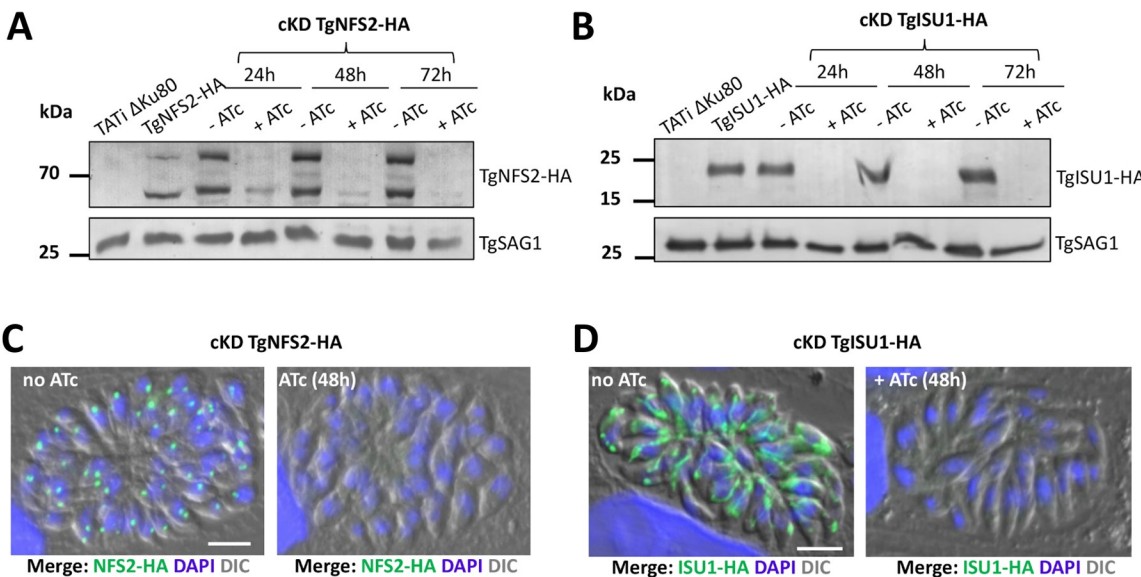

**Fig 3. Efficient down-regulation of TgNFS2 and TgISU1 expression with anhydrotetracyclin (ATc).** A) Immunoblot analysis with anti-HA antibody shows efficient down-regulation of TgNFS2 after 48h of incubation with ATc. Anti-SAG1 antibody was used as a loading control. B) Immunoblot analysis with anti-HA antibody shows efficient down-regulation of TgISU1 after 24h of incubation with ATc. Anti-SAG1 antibody was used as a loading control. C) and D) Immunofluorescence assays show TgNFS2 and TgISU1 are not detectable anymore after 48h of incubation with ATc. Scale bar represents 5 μm. DNA was labelled with DAPI. DIC: differential interference contrast.

absence or continuous presence of ATc for 7 days (Fig 4A and 4B). Depletion of both proteins completely prevented plaque formation, which was restored in the complemented cell lines. To assess whether this defect in the lytic cycle is due to a replication problem, all cell lines were preincubated in absence or presence of ATc for 48 hours and released mechanically, before infecting new host cells and growing them for an additional 24 hours in ATc prior to parasite counting. We noted that incubation with ATc led to an accumulation of vacuoles with fewer parasites, yet that was not the case in the complemented cell lines (Fig 4C and 4D). Overall, these data show that either TgNFS2 or TgISU1 depletion impacts parasite growth.

Then, we sought to assess if the viability of the mutant parasites was irreversibly affected. We thus performed another series of plaque assays, but at the end of the 7-day incubation, we washed out the ATc, incubated the parasites for an extra 7 days in the absence of ATc and evaluated plaque formation (Fig 4E). In these conditions, cKD TgNFS2-HA parasites displayed very small plaques suggesting their viability was irreversibly impacted. In contrast, cKD TgISU1-HA parasites showed plaques, suggesting parasite growth had at least partly resumed after ATc washout, while host cell lysis remained limited if the drug was kept continuously during the same period of time. This suggests that although depletion of TgISU1 has a marked impact on parasite growth, it is not completely lethal.

We performed IFAs to assess possible morphological defects that may explain the impaired growths of cKD TgNFS2-HA and cKD TgISU1-HA parasites. We stained the apicoplast and mitochondrion of parasites kept in the continuous presence of ATc for several days. cKD TgNFS2-HA parasites managed to grow and egress after three days and were seeded onto new host cells, where there were kept for two more days in the presence of ATc. During this second phase of intracellular development, and in accordance with the replication assays (Fig 4C), growth was slowed down considerably. Strikingly, while the mitochondrial network seemed

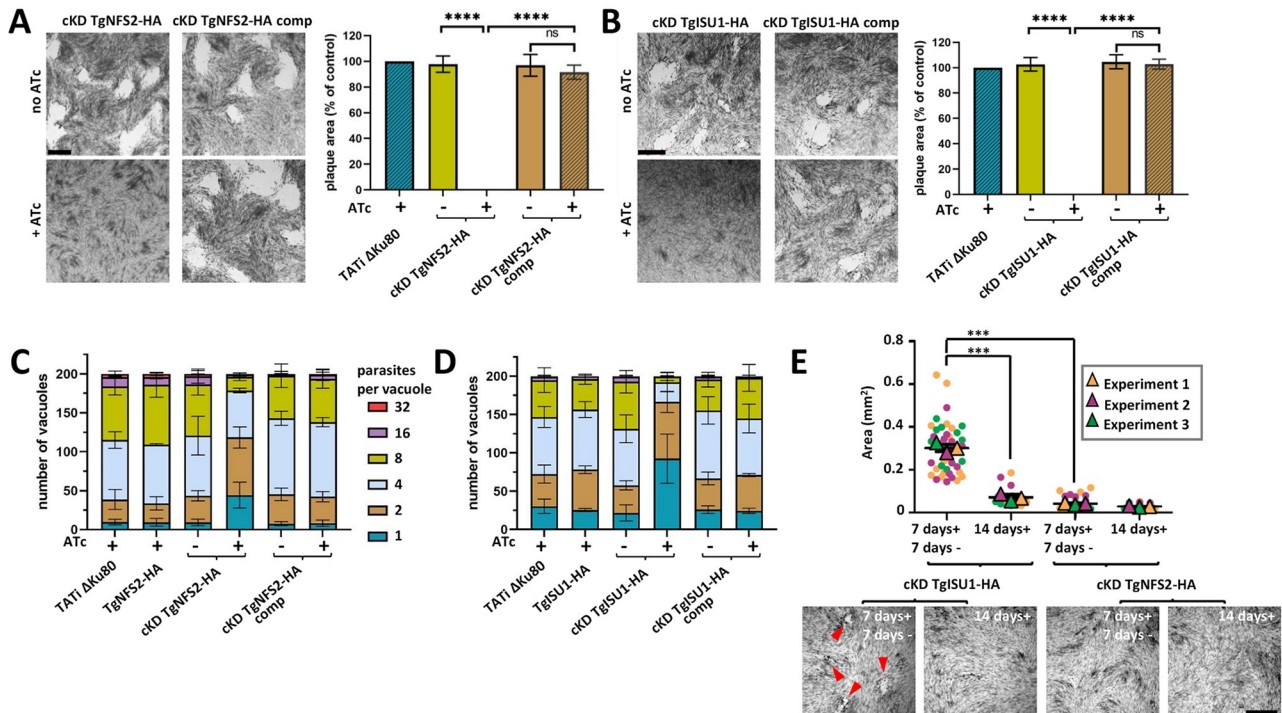

**Fig 4. Depletion of TgNFS2 or TgISU1 affects in vitro growth of the tachyzoites.** Plaque assays were carried out by infecting HFF monolayers with the TATi ΔKu80 cell line, the cKD TgNFS2-HA (A) or the cKD TgISU1-HA (B) cell lines, or parasites complemented with a wild-type version of the respective proteins. They were grown for 7 days ± ATc. Measurements of lysis plaque areas are shown on the right and highlight a significant defect in the lytic cycle when TgNFS2 (A) or TgISU1 (B) were depleted. Values are means of $n$ = 3 experiments ± SEM. Mean value of TATi ΔKu80 control was set to 100% as a reference. **** denotes $p \leq 0.0001$, Student's $t$-test. Scale bars = 1mm. TgNFS2 (C) and TgISU1 (D) mutant and complemented cell lines, as well as their parental cell lines and the TATi ΔKu80 control, were grown in HFF in the presence or absence of ATc for 48 hours, and subsequently allowed to invade and grow in new HFF cells for an extra 24 hours in the presence of ATc. Parasites per vacuole were then counted. Values are means ± SEM from $n$ = 3 independent experiments for which 200 vacuoles were counted for each condition. **E**) Plaque assays for the TgNFS2 and TgISU1 mutants were performed as described in A) and B), but ATc was washed out after 7 days (7 days+ 7 days-) or not (14 days+), and parasites were left to grow for an extra 7 days. Plaque area was measured. Data are means ± SEM from three independent experiments. *** $p \leq 0.001$, Student's $t$-test. Arrowheads show plaques forming in the TgISU1 upon ATc removal. Scale bar = 1mm.

normal, we noticed a progressive loss of the apicoplast marker TgCPN60 (Fig 5A), which was quantified (Fig 5B). As this could reflect a specific impact on this protein marker rather than a loss of the organelle, we also stained the parasites with fluorescent streptavidin, which mainly detects the biotinylated apicoplast protein acetyl-CoA carboxylase [49], confirming a similar loss of signal (S6 Fig). This suggests there is a general impact of TgNFS2 depletion on the organelle. Moreover, the fact that TgNFS2-depleted parasites eventually managed to perform a first lytic cycle and reinvade host cells before being blocked (Fig 5A) is reminiscent to the "delayed death" effect observed in apicoplast-defective parasites [6,50,51]. On the other hand, we were able to grow cKD TgISU1-HA parasites for five days of continuous culture: they developed large vacuoles and showed little sign of egress from the host cells (Fig 5C). Both the mitochondrion and the apicoplast appeared otherwise normal morphologically. These large vacuoles could reflect a defect in the parasite egress stage of the lytic cycle [52]. We thus performed an egress assay on cKD TgISU1-HA parasites that were kept for up to five days in the presence of ATc, and they were able to egress normally upon addition of a calcium ionophore (Fig 5D). These large vacuoles are also reminiscent of cyst-like structures [53], so alternatively this may reflect spontaneous stage conversion. Cysts are intracellular structures that contain the slow-growing form of *T. gondii*, called the bradyzoite stage (which is responsible for the

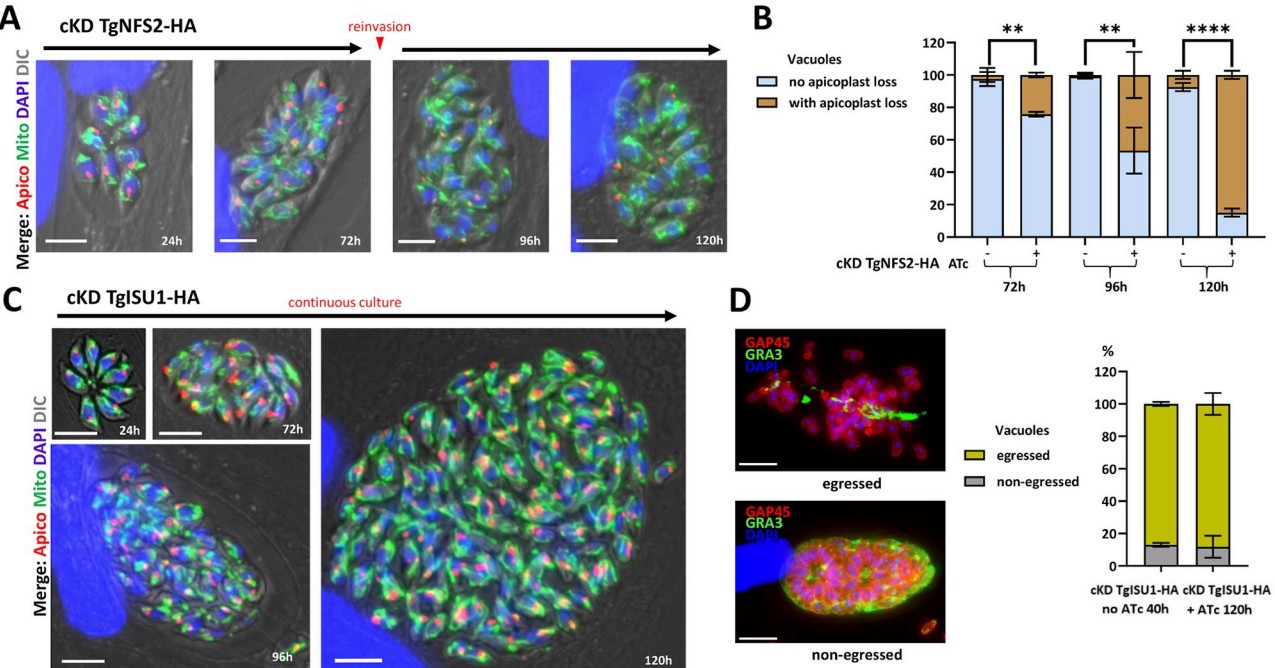

**Fig 5. Impact of TgNFS2 and TgISU1 depletion on intracellular tachyzoites.** A) Depletion of TgNFS2 impacts the apicoplast. cKD TgNFS2-HA parasites were kept in the presence of ATc and the aspect of the apicoplast and mitochondrion was evaluated by microscopic observation using specific markers (CPN60 and F1β ATPase, respectively). After 72 hours, parasites egressed and were used to reinvade new host cells for subsequent timepoints. Scale bar represents 5 μm. DNA was labelled with DAPI. DIC: differential interference contrast. B) Quantification of apicoplast loss in vacuoles containing cKD TgNFS2-HA parasites after 72 to 120 hours of incubation with ATc. Data are mean values from $n$ = 3 independent experiments ±SEM. ** $p \leq 0.01$, **** $p \leq 0.0001$, Student's $t$-test. C) Depletion of TgISU1 does not impact mitochondrial and overall parasite morphologies, but affects parasite growth. cKD TgISU1-HA parasites were grown in the presence of ATc for up to five days and the aspect of the apicoplast and mitochondrion was evaluated by microscopic observation using specific markers described in A). Growth in the presence of ATc was continuous for up to five days. Scale bar represents 5 μm. DNA was labelled with DAPI. DIC: differential interference contrast. D) Egress is not affected by TgISU1depletion. An egress assay was performed using calcium ionophore A23187. On the left are representative images of vacuoles containing parasites that egressed normally or did not. GRA3 (parasitophorous vacuole marker) staining is shown in green and GAP45 (parasite periphery marker) in red. Scale bars = 10 μm. On the right is the quantification of egress for cKD TgISU1-HA parasites kept in the presence of ATc or not. Mean values ± SEM from $n$ = 3 independent biological experiments are represented.

chronic phase of the disease), and they may appear even during in vitro growth in particular stress conditions [54]. Yet, this mutant cell line was generated in a type I *T. gondii* strain, which is associated with acute toxoplasmosis in the mouse model [55], and typically does not spontaneously form cysts. So, to be confirmed this hypothesis needed further investigations, as we will see later in the manuscript.

In any case, our data show that interfering with the plastidic and mitochondrial Fe-S protein pathways both had important consequences on parasite growth, but had a markedly different impact at a cellular level.

## Use of label-free quantitative proteomics to identify pathways affected by TgNFS2 or TgISU1 depletion

There is a wide variety of eukaryotic cellular processes that are depending on Fe-S cluster proteins. To get an overview of the potential *T. gondii* Fe-S proteome, we used a computational tool able to predict metal-binding sites in protein sequences [56], and performed subsequent manual curation to refine the annotation. We identified 64 proteins encompassing various cellular functions or metabolic pathways that included, beyond the Fe-S synthesis machinery

itself, several DNA and RNA polymerases, proteins involved in redox control and electron transfer, and radical S-adenosylmethionine (SAM) enzymes involved in methylation and methylthiolation (S2 Table). HyperLOPIT data or manual curation helped us assign a putative localization for these candidates. A considerable proportion (19%) of these were predicted to localize to the nucleus, where many eukaryotic Fe-S proteins are known to be involved in DNA replication and repair [57]. Yet, strikingly, most of the predicted Fe-S proteins likely localize to the endosymbiotic organelles. Several (19%) are predicted to be apicoplast-resident proteins, including radical SAM enzymes lipoate synthase (LipA) [58] and MiaB, a tRNA modification enzyme [59], as well as the IspG and IspH oxidoreductases of the non-mevalonate isoprenoid pathway [60]. Finally, for the most part (43%), candidate Fe-S proteins were predicted to be mitochondrial, with noticeably several important proteins of the respiratory chain (SDHB, the Fe-S subunit of the succinate dehydrogenase complex, Rieske protein and TgApi-Cox13) [61–63], but also enzymes involved in other metabolic pathways such as heme or molybdopterin synthesis. CRISPR/Cas9 fitness scores [39] confirmed many of these putative Fe-S proteins likely support essential functions for parasite growth.

We sought to confirm these results experimentally. Thus, in order to uncover the pathways primarily affected by the depletion of TgISU1 and TgNFS2, and to identify potential Fe-S protein targets, we conducted global label-free quantitative proteomic analyses. Like most plastidic or mitochondrial proteins, candidate Fe-S acceptors residing in these organelles are nuclear-encoded and thus need to be imported after translation and have to be unfolded to reach the stroma of the organelle. This not only implies the addition of the Fe-S cofactor should happen locally in the organelle, but also that this may have a role in proper folding of these proteins. We thus assumed that disrupting a specific pathway may have a direct effect on the stability and expression levels of local Fe-S proteins. Cellular downstream pathways or functions may also be affected, while other pathways may be upregulated in compensation. Parasites were treated for two days with ATc (cKD TgISU1-HA) or three days (cKD TgNFS2-HA, as it takes slightly longer to be depleted, Fig 3A), prior to a global proteomic analysis comparing protein expression with the ATc-treated TATi ΔKu80 control. For each mutant, we selected candidates with a log2(fold change) ≤-0.55 or ≥0.55 (corresponding to a ~1.47-fold change in decreased or increased expression) and a p-value <0.05 (ANOVA, $n$ = 4 biological replicates) (S3 and S4 Tables and Fig 6A and 6B). To get a more exhaustive overview of proteins whose amounts varied drastically, we completed this dataset by selecting some candidates that were consistently and specifically absent from the mutant cell lines or only expressed in these (S3 and S4 Tables).

Overall, depletion of TgISU1 led to a higher variability in protein expression and while the pattern of expression was essentially specific for the respective mutants, a number of shared variant proteins were found (Fig 6C and S5 Table). For instance, common lower expressed candidates include a SAM synthase, possibly reflecting a general perturbation of SAM biosynthesis upon loss of function of Fe-S-containing radical SAM enzymes [64]. Using dedicated expression data [65,66] available on ToxoDB.org we realized that, strikingly, many of the common variant proteins were stage-specific proteins (S5 Table). For instance, the protein whose expression went down the most is SAG-related sequence (SRS) 19F. The SRS family contains GPI-anchored surface antigens related to SAG1, the first characterized *T. gondii* surface antigen, and whose expression is largely stage-specific [67]. This protein, SRS19F, may be most highly expressed in stages present in the definitive host [66,68]. Conversely, SRS44, also known as CST1 and one of the earliest marker of stage conversion to bradyzoites [69], was upregulated in both mutants. Several other bradyzoite proteins whose expression increased included Ank1, a tetratricopeptide-repeat protein highly upregulated in the cyst-stages but not necessary for stage conversion [70], aspartyl protease ASP1, an α-galactosidase, as well as several dense granule proteins (GRA). Dense granules are specialized organelles that secrete GRA

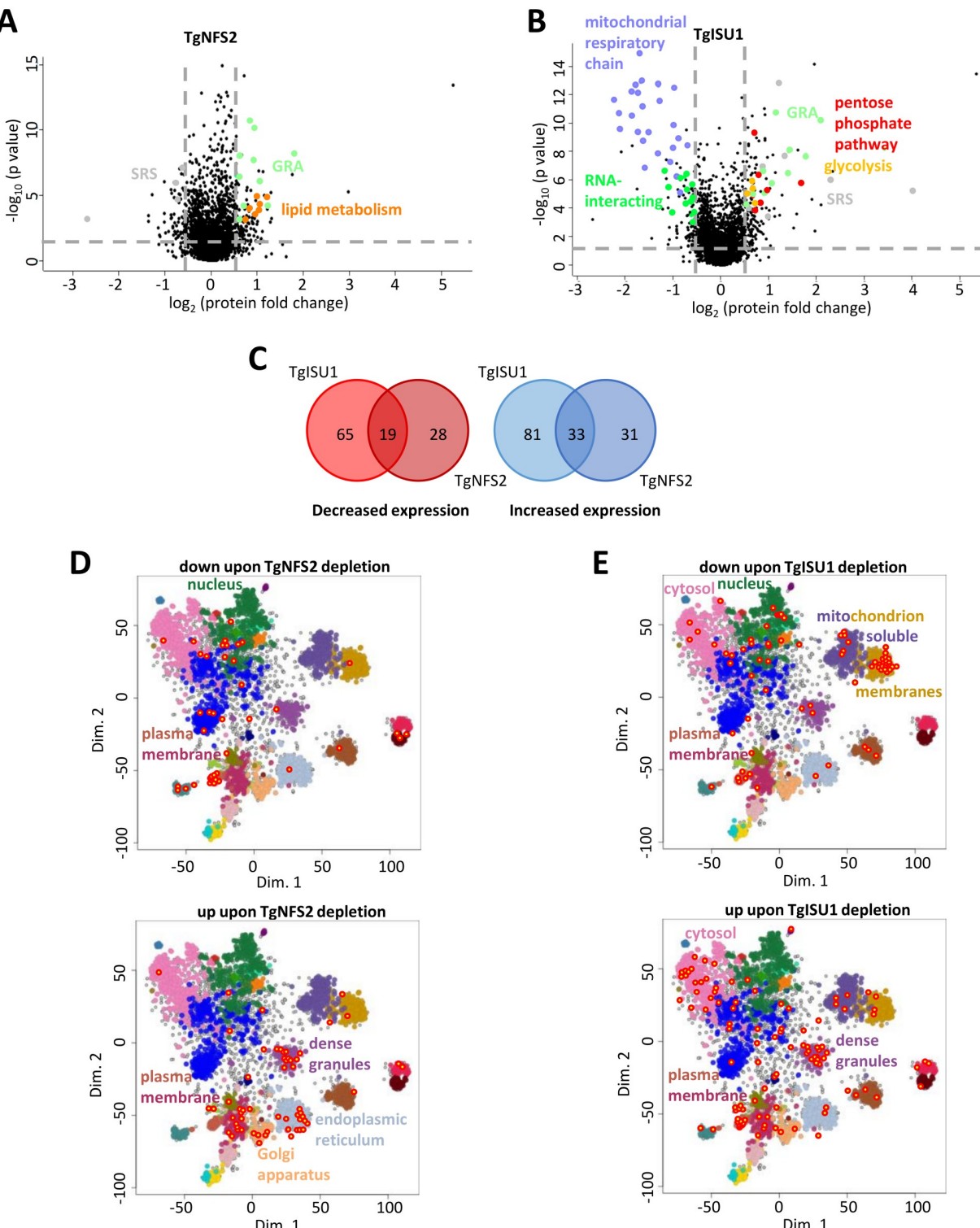

**Fig 6. Change in protein expression induced by TgNFS2 and TgISU1 depletion.** Volcano plots showing the protein expression difference based on label-free quantitative proteomic data from TgNFS2 (A) and TgISU1 (B) mutants grown in the presence of ATc. X-axis shows log2 fold change versus the TATi ΔKu80 control grown in the same conditions, and the Y-axis shows -log10(p value) after ANOVA statistical test for *n* = 4 independent biological replicates. Selected variant protein categories are highlighted in color. C) Venn diagram representation of the shared and unique proteins whose expression is affected by the depletion of TgNFS2 and TgISU1. D) and E): mapping of less or more abundant candidates (circled dots) in the TgNFS2 and TgISU1 mutants, respectively, on the spatial proteome map representation of the hyperLOPIT data, highlighting clusters denoting putative subcellular localization. Full details available at: https://proteome.shinyapps.io/toxolopittzex/.

proteins that are known to participate in nutrient acquisition, immune evasion, and host cell-cycle manipulation. Many GRA have been characterized in the tachyzoite stage, but several are stage-specific and expressed in bradyzoites [71]. It should be noted that bradyzoite-specific proteins were generally much strongly expressed upon TgISU1 depletion than TgNFS2 depletion. Nevertheless, altogether these results show that altering either the plastidic or the mitochondrial Fe-S cluster synthesis pathway led to an initial activation of the expression of some markers of the bradyzoite stage, whose involvement in the stress-mediated response is well documented [54].

## Depletion of TgNFS2 has an impact on the apicoplast, but also beyond the organelle

We next focused on proteins that varied specifically upon depletion of TgNFS2 (S3 Table). Using the hyperLOPIT data available on ToxoDB.org, we assessed the putative localization of the candidates (Fig 6D and S7A Fig) and we also defined putative functional classes based on manual curation (Fig 6A and S7B Fig). Surprisingly, few apicoplast proteins were impacted. This could reflect a limited impact on apicoplast Fe-S apoproteins, but this is in contradiction with the late, yet pronounced, effect we see on the organelle in the absence of TgNFS2 (Fig 5A and 5B). There might also be a bias due to an overall low protein abundance: less than half of the apicoplast candidates of the predicted Fe-S proteome (S2 Table) were robustly detected even in the control for instance, including our target protein TgNFS2. Finally, of course it is possible that depletion of Fe-S clusters, while impacting the functionality of target proteins, did not have a considerable effect on their abundance. We sought to verify this for apicoplast stroma-localized LipA, a well-established and evolutionarily-conserved Fe-S cluster protein, which was found to be only marginally less expressed in our analysis (S3 Table). LipA is responsible for the lipoylation of a single apicoplast target protein, the E2 subunit of the pyruvate dehydrogenase (PDH) [33]. Using an anti-lipoic acid antibody on cKD TgNFS2-HA protein extracts, we could already see a marked decrease in lipoylated PDH-E2 after only one day of ATc incubation (Fig 7A). This was not due to a general demise of the apicoplast as it considerably earlier than the observed loss of the organelle (Fig 5A and 5B), and levels of the CPN60 apicoplast marker were clearly not as markedly impacted (Fig 7A). This is also unlikely to reflect a general decrease of TgPDH-E2 levels upon TgNFS2 knock-down, as our quantitative proteomics data, which was performed after 3 days of ATc incubation, show the same amount of unique peptides for TgPDH-E2 (32 ± 2.5 for the control and 32.25 ± 2.75 for the mutant, ~60% of sequence coverage). This finding suggests apicoplast Fe-S-dependent activities may be specifically affected in this mutant, which would happen before observing the general demise and loss of the organelle. Long term incubation of cKD TgNFS2-HA parasites with ATc and co-staining with apicoplast and inner membrane complex (IMC) markers, revealed general cell division defects, including organelle segregation problems and an abnormal membranous structures (Fig 7B). Overall, this suggests impacting the Fe-S cluster synthesis pathway in the apicoplast had important consequences beyond the organelle itself.

## Depletion of TgISU1 impacts the mitochondrial respiratory chain

We also analyzed the proteins whose abundance changed upon TgISU1 depletion (S4 Table). Again, we used hyperLOPIT data to determine the localization of variant proteins (Fig 6E and S8A Fig) and we also inferred their potential function from GO terms or manual curation (Fig 6B and S8B Fig). Depletion of TgISU1 had a notable impact locally, as numerous mitochondrial proteins were found in lower abundance. Remarkably, most of these proteins were identified as members of the mitochondrial respiratory chain: while our proteomic study detected

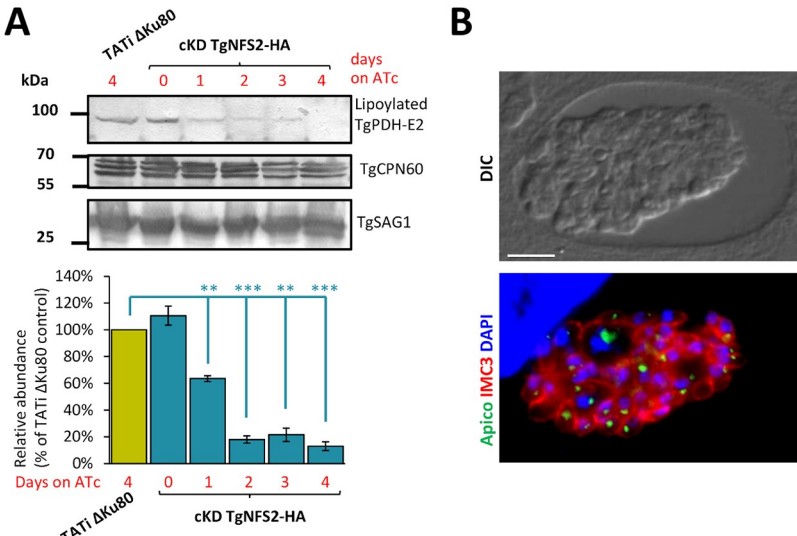

**Fig 7. TgNFS2 depletion impacts apicoplast-related pathways and has deleterious effects on parasite replication.**
A) A decrease in the lipoylation of the E2 subunit of pyruvate dehydrogenase (TgPDH-E2), which depends on the Fe-S-containing lipoyl synthase called LipA in the apicoplast, was observed by immunoblot using an anti-lipoic acid antibody on cell extracts from cKD TgNFS2-HA parasites kept with ATc for an increasing period of time. TgCPN60 was used as a control for apicoplast integrity. TgSAG1 was used as a loading control. Decrease of lipoylated TgPDH-E2 was quantified by band densitometry and normalized with the internal loading control. Data represented are mean ±SEM of $n = 3$ independent experiments. ** $p \leq 0.01$, *** $p \leq 0.001$ ANOVA comparison. B) cKD TgNFS2-HA parasites that were grown in the presence of ATc for 5 days were co-stained with anti-TgIMC3 (to outline parasites and internal buds) and anti-CPN60 (an apicoplast marker), which highlighted abnormal membrane structures and organelle segregation problems. Scale bar represents 5 μm. DNA was labelled with DAPI. DIC: differential interference contrast.

peptides corresponding to as much as 92% of the hyperLOPIT-predicted mitochondrial membrane and soluble proteins, 12% of these were found to be significantly less expressed upon TgISU1 depletion, 60% of which are known mitochondrial ETC components (Figs 6A and 8A and S4 Table). This suggests a specific effect of TgISU1 depletion on the mitochondrial ETC. This ETC comprises four complexes in Apicomplexa (which typically lack complex I), in which several Fe-S proteins have important function. As mentioned earlier, they include the Fe-S subunit of the succinate dehydrogenase complex (SDHB, part of complex II), the Rieske protein (part of complex III) and TgApiCox13 (part of complex IV) [61–63]. Not only these three Fe-S cluster proteins were found to be less expressed upon TgISU1 depletion, but about 60% and 80% of complexes III and IV components (including recently characterized parasite-specific subunits [62,63]), respectively, were also significantly less abundant (S4 Table and Fig 8A). The impact on components of complexes III and IV beyond their respective Fe-S-dependent subunits is not surprising: as shown by others in *T. gondii* depletion of selected members of a mitochondrial ETC complex can result in stalled assembly or impaired stability of the whole complex [61–63,72].

We sought to verify the impact of TgISU1 depletion on proteins of the mitochondrial respiratory chain by tagging two candidates, TgSDHB and TgApiCox13. In order to do this, due to the lack of efficient selectable marker in the cKD TgISU1-HA cell line, we first had to generate a new independent cKD TgISU1 untagged mutant from the TATi ΔKu80 cell line (S9A and S9B Fig). In this cell line, we verified proper regulation of *TgISU1* expression by ATc (S9C Fig) and impact of TgISU1 depletion on parasite growth (S9D Fig). In this cKD TgISU1 mutant, a sequence coding for a C-terminal triple HA epitope tag was inserted at the endogenous

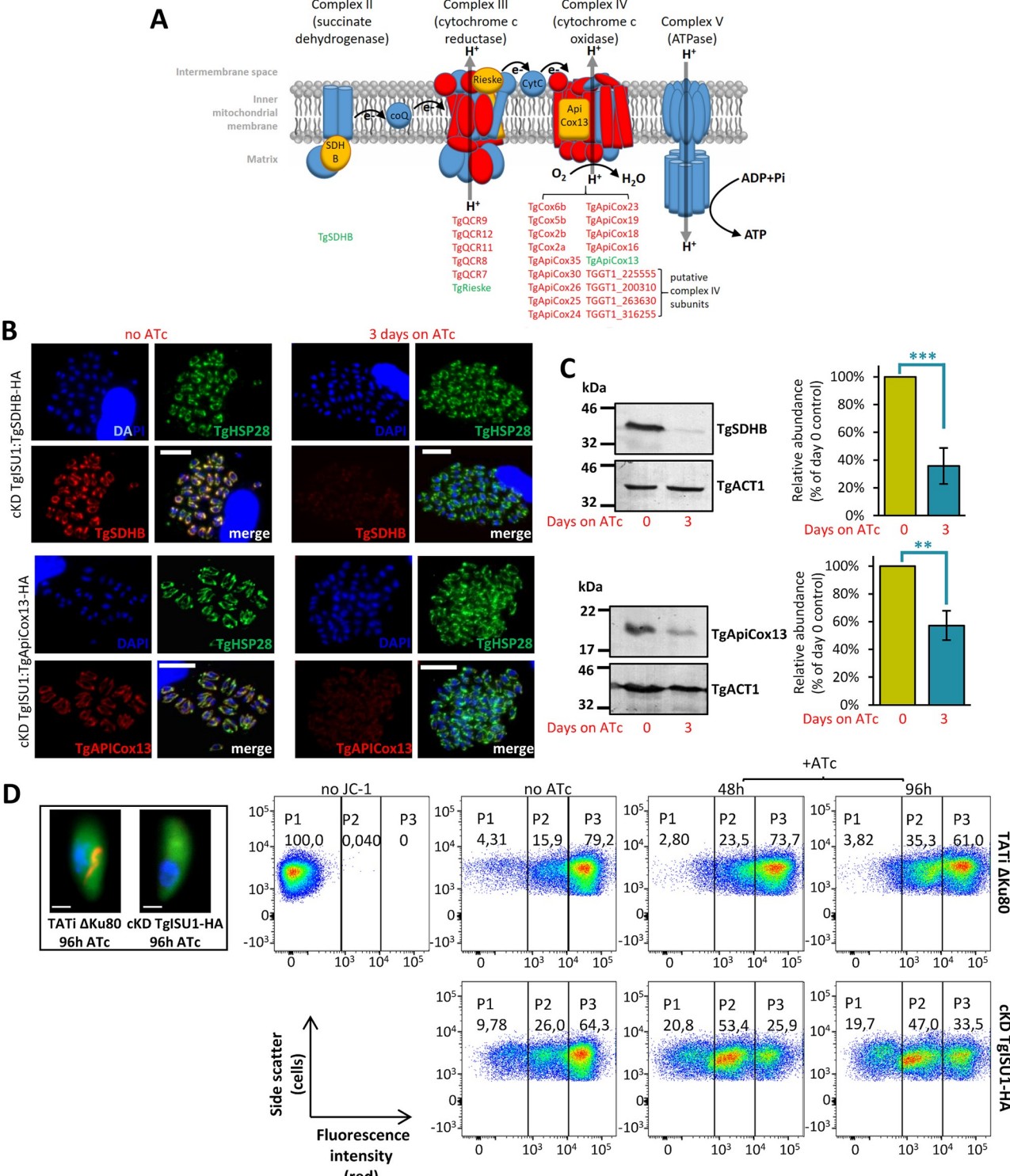

**Fig 8. TgISU1 depletion impacts the mitochondrial respiratory chain.** A) Schematic representation of the *T. gondii* mitochondrial respiratory chain; listed are the subunits of the different complexes that were found less abundant upon TgISU1 depletion; Fe-S proteins are highlighted in green. SDH: succinate dehydrogenase; CoQ: coenzyme Q B) Immunofluorescence analysis of cKD TgISU1 parasites expressing HA-tagged TgSDHB (top) or TgApiCox13 (bottom) showing a decrease in expression in the HA-tagged candidates after 3 days of incubation with ATc. Images were taken with the same exposure time for similar channels. TgHSP29 was used as a mitochondrial marker. Scale bar represents 10 μm. DNA was labelled with DAPI. C) Immunoblot analysis of TgSDHB (top) and TgApiCox13 (bottom) levels upon depletion of TgISU1 after 3 days of incubation with ATc. TgACT1 was

used as a loading control. Protein levels were quantified by band densitometry and normalized with the internal loading control. Data represented are mean ±SEM of $n$ = 5 independent experiments. ** $p \leq 0.01$, *** $p \leq 0.001$, ANOVA comparison. D) Impact of TgISU1 depletion on the parasite mitochondrial membrane potential was measured by JC-1 labelling. cKD TgISU1-HA parasites were grown or not in the presence of ATc, mechanically released from their host cells and labelled with the JC-1 dye. This dye exhibits potential-dependent accumulation in the mitochondrion, indicated by a switch from green fluorescence for the monomeric form of the probe, to a concentration-dependent formation of red aggregates (inset, DNA is labelled with DAPI and shown in blue, scale = 1μm). Parasites were then analysed by flow cytometry. Unlabelled parasites (no JC-1) was used as a control for gating. Numbers represent the percentage of cells in each of the subpopulations (P1, P2, P3). One representative experiment out of $n$ = 3 biological replicates is shown.

*TgSDHB* or *TgApiCox13* locus by homologous recombination (S10 Fig). We then incubated these parasites with ATc for three days and used an anti-HA antibody to detect and quantify the proteins of interest by IFA (Fig 8B) and immunoblot (Fig 8C). In accordance with the quantitative proteomics data, both TgSDHB and TgApiCox13 were found to be less expressed in absence of TgISU1.

This suggested the mitochondrial membrane potential and consequently the respiratory capacity of the mitochondrion were likely altered in the absence of a functional mitochondrial Fe-S cluster synthesis pathway. To verify this, we performed flow cytometry quantification using JC-1, a monomeric green fluorescent carbocyanine dye that accumulates as a red fluorescent aggregates in mitochondria depending on their membrane potential (Fig 8D). Depletion of TgISU1 led to a marked decrease of the parasite population displaying a strong red signal (Fig 8D). The effect was maximal after two days of ATc treatment and not further increased by a four-day treatment, which is consistent with the quantitative proteomics data already showing strong impact on proteins from complexes II, III and IV after only two days of ATc treatment. It should be noted that although we believe the drop in mitochondrial membrane potential is likely due to a specific alteration of the respiratory chain, it may also be due to a loss of parasite viability. Yet, reassuringly our results are in line with the recent findings obtained by Aw et al., who generated a mutant of mitochondrial Fe-S cluster synthesis (by depleting TgNFS1), and observed a sharp decrease in TgSDHB abundance and a clear drop in mitochondrial $O_2$ consumption rate [25].

Concomitantly to the lesser expression of mitochondrial respiratory chain subunits, the proteomics analysis revealed TgISU1 depletion induced a significant increase in cytosolic enzymes involved in glycolysis, as well as its branching off pentose phosphate pathway (Fig 8A and 8B and S4 Table). The upregulation of glycolytic enzymes potentially reflects a metabolic compensation for mitochondrial defects in energy production due to the impairment of the respiratory chain. Other proteins whose abundance was markedly decreased were predicted to be cytoplasmic or nuclear, including proteins involved in DNA repair and replication (S4 Table), which is perhaps unsurprising as the cytosolic CIA Fe-S cluster assembly pathway is supposedly dependent from the ISC pathway [24]. Finally, the changes in abundance of several RNA-binding proteins involved in mRNA half-life or transcription/translation regulation may also reflect adaptation to a stress (S4 Table).

## Depletion of TgISU1 initiates conversion into bradyzoites

Another feature highlighted by the quantitative proteomics analysis of the TgISU1 mutant is the change in the expression of stage-specific proteins (S4 Table). The expression of several bradyzoite-specific proteins including GRAs and proteins of the SRS family, was strongly increased. At the same time, some tachyzoite-specific SRS and GRA proteins were found to be less expressed. This was supporting the idea that intracellularly developing parasites lacking TgISU1 may convert into bona fide cyst-contained bradyzoites, as suggested by our initial morphological observations (Fig 5C). To verify this experimentally, we used a lectin from the

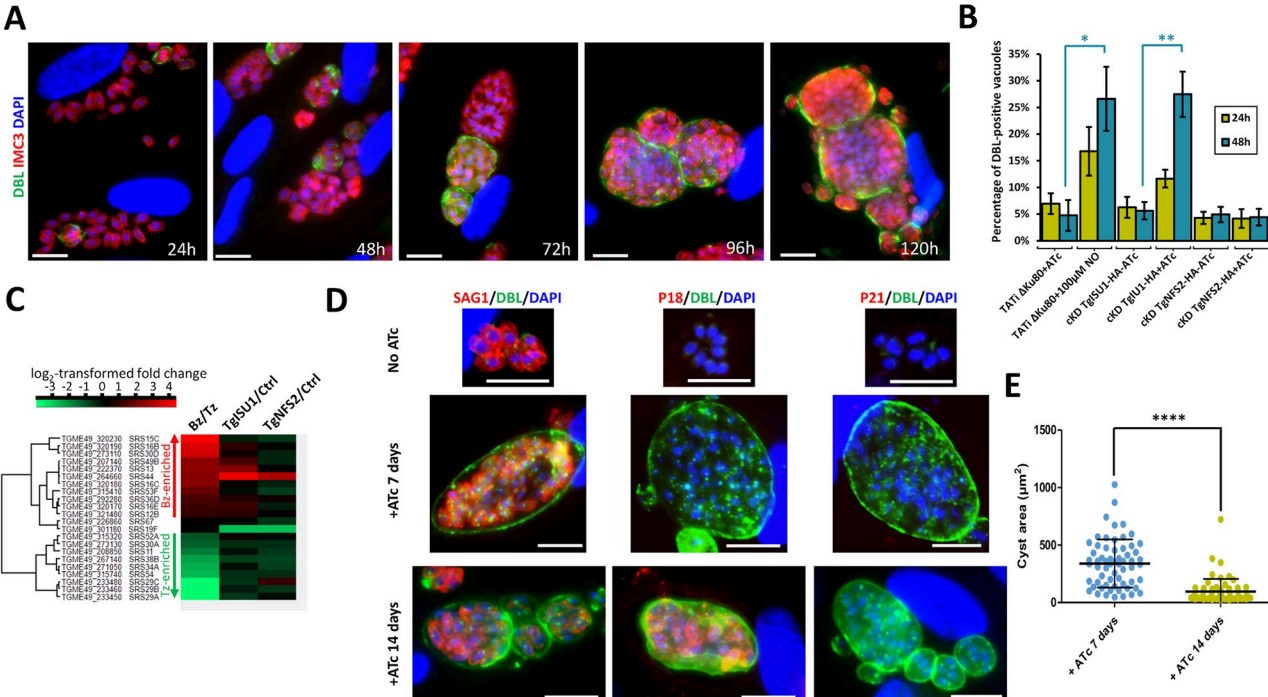

**Fig 9. Depletion of TgISU1 triggers parasite differentiation.** A) cKD TgISU1-HA parasites were grown in the presence of ATc and labelled with anti-TgIMC3 (to outline parasites) and a lectin of *Dolicos biflorus* (DBL) to specifically outline nascent cyst walls. Scale bar represents 10 μm. DNA was labelled with DAPI. DIC: differential interference contrast. B) Quantification of DBL-positive vacuoles after 24 hours or 48 hours of culture of 1) the cKD TgISU1-HA and cKD TgNFS2-HA mutants in the presence or absence of ATc 2) the Tati ΔKu80 cell line, as a negative control, 3) the Tati ΔKu80 cell line in the presence of 100μM nitric oxide (NO), as a positive control. Data are from *n* = 3 independent experiments. Values are mean ±SEM. * *p* ≤ 0.05, ** *p* ≤ 0.01, Student's *t*-test C) Clustering of bradyzoite (Bz) or tachyzoite (Tz)-specific proteins of the SRS family shows specific enrichment of bradyzoite proteins upon TgISU1 depletion. D) The cKD TgISU1-HA mutant was grown for up to 14 days in the presence of ATc and co-stained with early cyst wall marker DBL together with tachyzoite marker SAG1, or intermediate (P18/SAG4), or late (P21) bradyzoite markers. Scale bar represents 10 μm. DNA was labelled with DAPI. E) Measurement of the cyst area size after growing the cKD TgISU1-HA mutant for 7 and 14 days in the presence of ATc and labelling the cyst wall with DBL and measuring the surface of 60 cysts per condition. Mean ±SD is represented. One representative experiment out of three independent biological replicates is shown. **** denotes *p* ≤ 0.0001, Student's *t*-test.

plant *Dolichos biflorus* (DBL), which recognizes the SRS44/CST1 glycoprotein that is exported to the nascent wall of differentiating cysts [69]. We could see that during continuous growth of cKD TgISU1-HA parasites in the presence of ATc, there was an increasing number of DBL-positive structures (Fig 9A). This was quantified during the first 48 hours of intracellular development (Fig 9B) and, interestingly, was shown to mimic the differentiation induced by nitric oxide, a known factor of stage conversion [73], and a potent damaging agent of Fe-S clusters [74]. We combined RNAseq expression data for tachyzoite and bradyzoite stages [66] to establish a hierarchical clustering of the SRS proteins detected in our quantitative proteomics experiments for the two mutants (Fig 9C). Overall, this revealed an increase in the expression of bradyzoite-specific SRS in the TgISU1 mutant, although not all bradyzoite-specific SRS were strongly increased, perhaps reflecting an atypical or incomplete stage conversion. As mentioned earlier, some were also upregulated in the TgNFS2 mutant but in much lesser proportions. The strongest increase in bradyzoite-specific SRS expression upon TgNFS2 depletion was for SRS44/CST1, which happens to be the protein DBL preferentially binds to [69]. However, contrarily to the TgISU1 mutant, labelling experiments did not indicate any detectable increase in DBL recruitment in the TgNFS2 mutant (Fig 9B), confirming that impairing the plastidic Fe-S center synthesis pathway does not trigger full stage conversion.

Stage conversion is a progressive process that happens over the course of several days, as it involves the expression of distinct transcriptomes and proteomes [54]. Markers for specific steps of in vitro cyst formation had been previously described [75], so we have used several of these to check the kinetics of stage conversion in the TgISU1-depleted parasites. We kept the cKD TgISU1-HA parasites for up to 14 days in the presence of ATc and tested for the presence of SAG1 (tachyzoite maker), DBL (early bradyzoite marker), P18/SAG4 (intermediate bradyzoite marker) and P21 (late bradyzoite marker) (Fig 9D and S11B Fig). After 7 days of ATc treatment, the DBL-positive cyst contained parasites that were still expressing SAG1 and not yet P18/SAG4, whereas after 14 days parasites with P18/SAG4 labelling were found, but there was still a residual SAG1 expression; expression of late marker P21 was, however, never detected. As controls, the TATi ΔKu80 parental cell line (derived from the RH type I strain) and the Prugniaud cystogenic type II strain were subjected to alkaline stress-induced stage conversion [76] for a similar duration (S11A and S11B Fig). While a majority of DBL-positive cysts of the type II strain expressed the P18/SAG4 and P21 markers after 14 days of differentiation, this was not the case for the type I TATi ΔKu80 cell line. Cysts containing TgISU1-depleted parasites thus displayed a somewhat intermediate phenotype with the expression of P18/SAG4, but not P21. This suggests stage conversion of these parasites progresses beyond the appearance of early cyst wall markers, but it seems incomplete. In fact, DBL-positive cysts showed a marked decrease in their mean size between the 7 and 14 days timepoints upon TgISU1 depletion (Fig 9D and S11C Fig). Smaller cyst size seemed to be a feature of the parental TATi ΔKu80 cell line when compared with the type II strain (S11C Fig), and noticeably these type I parasites also kept growing largely as tachyzoites and were able to reinitiate invasion cycles, leading to considerable host cell lysis during the course of the pH stress-induced conversion. Decrease in cyst size over time may thus reflect incomplete conversion, and be a consequence of later reactivation/reinvasion events; in the case of the TgISU1 mutant, the markedly smaller cyst size after 14 days of protein depletion (Fig 9E and S11C Fig) may also suggest a lack of fitness in the long term.

## Lack of lethality and initiation of stage conversion are features shared by other mitochondrial mutants

As impairment of the mitochondrial ETC and stage conversion are the two main features observed for the TgISU1 mutant, it raised the possibility the former may be involved in triggering the latter. We thus sought to evaluate in more details viability and differentiation of other mitochondrial mutants. We used the ATc-regulatable cKD cell line for TgQCR11 [63], a complex III subunit found less abundant in absence of TgISU1 (Fig 8A), and which was found by others to be essential for mitochondrial respiration and parasite growth [62,63]. We also included in our study an ATc-regulatable cKD cell line for TgmS35, a mitoribosomal protein whose depletion impacts organelle morphology and function (including mitochondrial respiration), and overall parasite fitness [77].

Like for TgISU1 (Fig 4E), we assessed whether or not the fitness phenotype of these mutants was reversible by performing plaque assays in the presence of ATc for 7 days, and then monitored plaque formation for another 7 days upon drug removal (Fig 10A). The TgQCR11 mutant behaved very similarly to the TgISU1 mutant, with virtually no plaques formed when incubated with ATc, while it was able to reactivate the lytic cycle upon ATc washout. The TgmS35 mutant seemed less affected by protein depletion: not only it generated small plaques after 7 days in the presence of ATc, but whether or not the drug was washed out, prolonged incubation led to large lysis plaques in the host cells. Overall, this experiment showed that mitochondrial mutants, whether they are affected in a metabolic or more structural function,

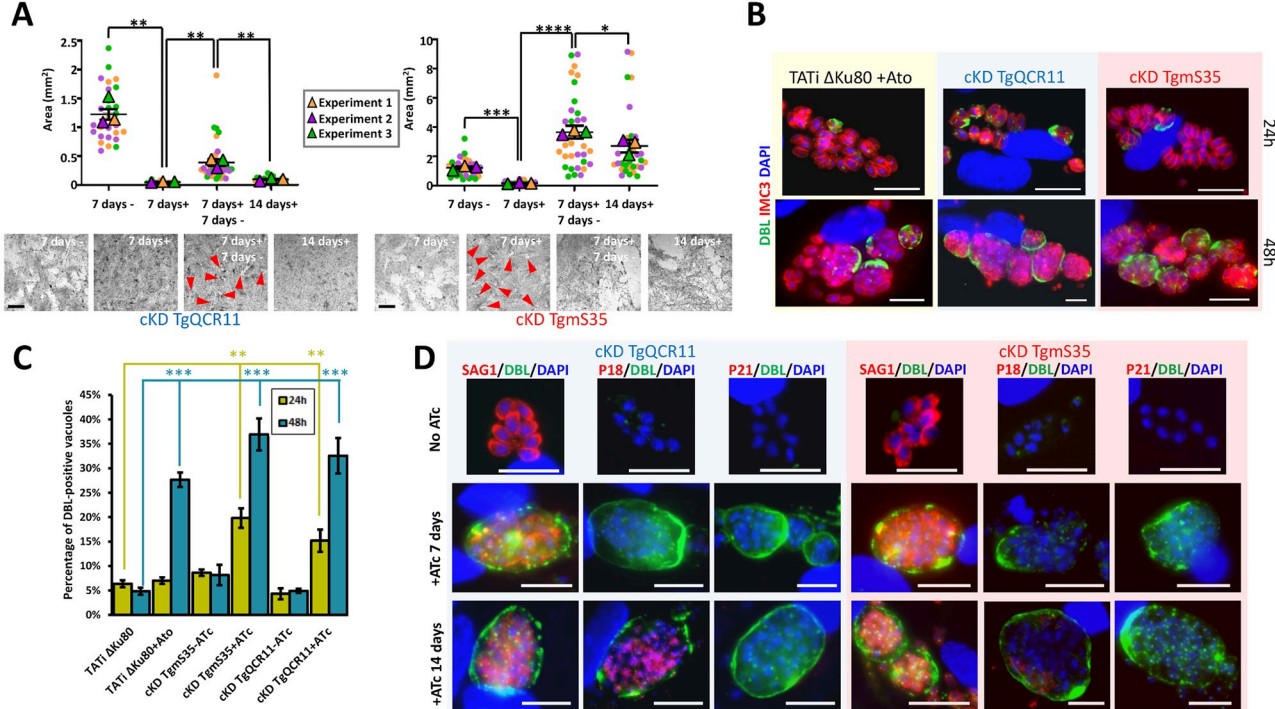

**Fig 10. Other mitochondrial mutants remain viable and initiate stage conversion.** A) Plaque assays for the TgQCR11 and TgmS35 mutants were performed by infecting HFF monolayers with cKD cell lines and letting them grow in absence (7 days -) or presence of ATc (7 days +), after 7 days ATc was washed out (7 days+ 7days-) or not (14 days+), and parasites were left to grow for an extra 7 days. Plaque area was measured. Data are means ± SEM from three independent experiments. * $p \leq 0.05$, ** $p \leq 0.01$, *** $p \leq 0.001$, **** $p \leq 0.0001$, Student's *t*-test. Arrowheads show small plaques. Scale bar = 1mm. B) Tati ΔKu80 parasites were grown in the presence of atovaquone and cKD TgQCR11 and TgmS35 parasites were grown in the presence of ATc for up to two days. They were labelled with anti-TgIMC3 (to outline parasites) and *Dolicos biflorus lectin* (DBL) to specifically outline nascent cyst walls. Scale bar represents 10 μm. DNA was labelled with DAPI. DIC: differential interference contrast. C) Quantification of DBL-positive vacuoles after 24 hours or 48 hours of culture of 1) the cKD TgQCR11 and cKD TgmS35 mutants in the presence or absence of ATc 2) the Tati ΔKu80 cell line in the presence or not of 1μM atovaquone. Data are from *n* = 3 independent experiments. Values are mean ±SEM. ** $p \leq 0.01$, *** $p \leq 0.001$, Student's *t*-test. D) The cKD TgQCR11 and cKD TgmS35 mutants were grown for up to 14 days in the presence of ATc and co-stained with early cyst wall marker DBL together with tachyzoite marker SAG1, or early (P18/SAG4), or late (P21) bradyzoite markers. Scale bar represents 10 μm. DNA was labelled with DAPI.

remain essentially viable. The similarity between the TgISU1 and the TgQCR11 mutant suggested the latter may be switching to a slow-growing, but still viable, bradyzoite stage. The slowed-down, but less hampered progression through the lytic cycle of the TgmS35 mutant suggested it might retain more tachyzoite-like growth kinetics.

To investigate the ability of these mutants to form cysts, we performed IFAs with specific bradyzoite markers, like we did previously for the cKd TgISU1-HA cell line (Fig 9A and 9B, S11B Fig). We first looked for the recruitment of DBL to parasite-containing vacuoles during the first 48 hours of protein depletion, (Fig 10B and 10C). As a control, we also treated TATi ΔKu80 parasites with atovaquone, an inhibitor of the mitochondrial ETC that was previously shown to trigger stage conversion [78]. Interestingly, like the atovaquone-treated TATi ΔKu80 parasites and the TgISU1 mutant (Fig 9A and 9B), treatment of both TgQCR11 and TgmS35 mutants with ATc seemed to trigger stage conversion, as illustrated by an increase in DBL staining of vacuoles over time (Fig 10B and 10C). When then tried to evaluate conversion over a longer period by keeping the parasites for up to 14 days in the presence of ATc (Fig 10D and S11B Fig). For the TgmS35 mutant, this was achieved by using very low doses of parasite inoculum, which allowed preserving some host cells in spite of the extensive lysis caused by the

parasites. Like for the TgISU1 mutant, even long term differentiation-inducing conditions did not allow complete disappearance of the SAG1 tachyzoite marker in the TgQCR11 and TgmS35 mutants, and staining of the late bradyzoites marker P21 was not observed. However, while a few DBL-labelled cysts remained in the TgmS35 culture after 14 days of ATc treatment, they very rarely showed staining with the intermediate bradyzoite marker P18/SAG4 (S11B Fig). On the contrary, substantial staining was observed with this marker upon long term depletion of TgQCR11, like we observed for the TgISU1 mutant (S11B Fig). Of note, as observed for the TgISU1 mutant and parental TATi ΔKu80 cell line, these other mutants generated in a type I background strain also displayed smaller cyst size than the type II parasites after long term induction of stage conversion (S11C Fig), suggesting reactivation may happen and incomplete differentiation is likely a general feature of RH-derived parasites.

In conclusion, our findings suggest that interfering with general function like mitochondrial translation, or targeting more specifically the mitochondrial ETC, does not irreversibly impair parasite viability and instead leads to an initiation of stage conversion into bradyzoites, although it may not necessarily be complete.

## Discussion

Because of their origin and metabolic importance, the two apicomplexan endosymbiotic organelles have gathered considerable interest as potential drug targets [79,80]. It may be obvious, as for example the plastid hosts several metabolic pathways which are not present in the mammalian hosts of these parasites. Yet, even for conserved housekeeping functions or, in the case of the mitochondrion early phylogenetic divergence, there may still be enough molecular differences to allow selective chemical inhibition. In fact, several drugs used for prophylactic or curative treatments against Apicomplexa-caused diseases are already targeting these organelles [81]. They are essentially impacting the organellar protein synthesis by acting on the translation machinery [82], although the mitochondrial ETC inhibitor atovaquone is also used to treat malaria and toxoplasmosis [83]. One main difference when targeting *Plasmodium* and *Toxoplasma* by drugs is that the latter easily converts into the encysted bradyzoite resistance form. It has been known for some time that treatment of tachyzoites with mitochondrial inhibitors triggers stage conversion [73,78,84]. This may be efficient to counteract the acute phase of toxoplasmosis, but at the same time may favour persistence of the parasites in the host.

Here we characterized Fe-S cluster synthesis pathways which are very similar biochemically, but are located into two distinct endosymbiotic organelles, and whose inactivation has drastically different consequences for parasite fitness. Fe-S clusters are ancient, ubiquitous and fundamental to many cellular functions, but their synthesis by distinct biosynthetic pathways was inherited by plastids or by the mitochondrion through distinct bacterial ancestors, and have thus specialized into adding these cofactors to different client proteins [22]. A key function of Fe-S clusters, owing to their mid-range redox potential, is electron transfer and redox reactions, mainly as components the respiratory and photosynthetic ETCs. They also have important functions in stabilizing proteins, redox sensing, or catalysis through SAM enzymes. Several of these are not retained in Apicomplexa, whose plastid has lost its photosynthetic ability for example. Nevertheless, our prediction of the *T. gondii* Fe-S proteins repertoire suggests many key functions associated with the apicoplast or the mitochondrion are likely to be affected by a perturbation of Fe-S assembly (S2 Table).

For the apicoplast, these include lipoic acid or isoprenoid synthesis. Inactivation of the apicoplast-located TgNFS2 had a late but marked effect on the organelle itself, as it led ultimately to a partial loss of the apicoplast, which is consistent with the phenotype observed when disrupting the SUF pathway in *Plasmodium* [26]. IspG and IspH, which are key Fe-S-dependent

enzymes of the non-mevalonate isoprenoid synthesis pathway [60], were only found marginally less expressed in our quantitative analysis after TgNFS2 depletion. However, our proteomics dataset provided indirect clues that their function may be impacted. Isoprenoid synthesis is vital for *T. gondii* tachyzoites [34], and it has implication beyond the apicoplast, as prenylated proteins or isoprenoid precursors are involved in more general cellular processes including intracellular trafficking or mitochondrial respiration [85]. Isoprenoids are for instance important for synthesizing ubiquinone/coenzyme Q, and the single predicted mitochondrial candidate that was significantly less expressed upon TgNFS2 depletion is a putative UbiE/COQ5 methyltransferase, involved in synthesis of this co-factor [86]. Isoprenoids are also important for dolichol-derived protein glycosylation and glycosylphosphatidylinositol (GPI)-anchor biosynthesis, and interestingly the three putative rhoptry-localized candidates significantly less expressed in the TgNFS2 mutant (S3 Table) are potentially GPI-anchored and/or glycosylated, as predicted through sequence analysis by dedicated webservers [87–89]. Overall, this might be an indication that TgNFS2 depletion impacts isoprenoid synthesis in the apicoplast, which in turn would impact other metabolic pathways.

Impairing isoprenoid synthesis does not, however, necessarily lead to a loss of the organelle [26]. There may thus be another explanation for this phenotype. Interestingly, we could show that perturbing the SUF pathway, which is supposedly important for Fe-S-containing enzyme LipA, impacts the lipoylation of the E2 subunit of the apicoplast-located PDH (Fig 7A). The PDH complex catalyzes the production of acetyl-CoA, which is the first step of the FASII system, and perturbation of either the PDH or other steps of the FASII system leads to a loss of the organelle and severely impairs fitness of the parasites [34,90]. Interestingly, long term depletion of TgNFS2 leads to cell division problems, and affects membrane compartments such as the IMC (Fig 7B), which are defects previously observed in parasites where the FASII system has been genetically- or chemically-disrupted [91–93]. Our quantitative proteomic analysis shows potential compensatory mechanisms may be used by the parasites in response this early perturbation of the apicoplast lipid metabolism that precedes organelle loss. Tachyzoites are indeed known to be able to use exogenous lipid sources to adapt metabolically [90,92] and, interestingly, upon depletion of TgNFS2 we observed a pattern of overexpression for ER-located enzymes involved in the synthesis of several phospholipids and ceramides (Fig 6A and 6D and S3 Table). These lipids are usually synthesized in the ER from apicoplast-generated precursors, as both organelles cooperate for FA and phospholipid (PL) synthesis [94]. Yet, the ER-localized PL-synthesis machinery can also use FA scavenged from the host [95]. The increased expression of ER-localized lipid-related enzymes may thus reflect an increased synthesis, potentially from exogenous lipid precursors, in compensation for a defect in the apicoplast-localized machinery. In spite of this potential compensation mechanism, it seems the alteration of the SUF pathway in *T. gondii* has such a profound impact that it ultimately leads to the irreversible demise of the parasites (Fig 4E). It would be interesting to use recently described approaches like stable isotope labelling of lipid precursors combined to lipidomic analysis [92], to investigate in the SUF pathway mutant the potential changes in de novo synthesis of FA or in lipid scavenging from the host.

In the mitochondrion, important pathways potentially involving Fe-S proteins include the respiratory ETC, the TCA cycle, as well as molybdenum and heme synthesis (S2 Table). Accordingly, perhaps the most obvious consequence of disrupting the ISC pathway was the profound impact on the mitochondrial respiratory capacity, as evidenced experimentally by measuring the mitochondrial membrane potential (Fig 8D), and supported by quantitative analyses showing a clear drop in expression of many respiratory complex proteins (Fig 8A and 8C and S4 Table). This is also in line with the recent description of another *T. gondii* mitochondrial Fe-S cluster synthesis mutant that showed a marked alteration of its respiratory

capacity [25]. Although the mitochondrion, through the TCA cycle and the respiratory chain/ oxidative phosphorylation, contributes to energy production in tachyzoites [96], the glycolytic flux is also believed to be a major source of carbon and energy for these parasites [97]. Thus, rather coherently, as highlighted by our quantitative proteomic analysis, disruption of the ISC pathway led to a potential overexpression of glycolytic enzymes concurrently with the lower expression of mitochondrial ETC components (Fig 6B and S4 Table). The possible overexpression of enzymes of the pentose phosphate pathway, which is branching off from glycolysis and is providing redox equivalents and precursors for nucleotide and amino acid biosynthesis, is also potentially indicative of a higher use of glucose in these conditions. The metabolic changes encountered by ISC-deficient parasites do not cause their rapid demise, as they are able to initiate conversion to the bradyzoite stage, which has been suggested to rely essentially on glycolysis for energy production [98]. Our analysis of the viability of other mitochondrial mutants confirmed results obtained by others showing that the organelle, and in particular the ETC, seem important for parasite fitness [62,63,77]. Yet, perhaps because of their metabolic flexibility, mutant parasites seem to retain the ability to survive as tachyzoites (as shown here and in [62]), or initiate a switch to bradyzoites.

The inactivation of the ISC pathway likely has consequences on other important cellular housekeeping functions besides mitochondrial metabolism. In other eukaryotes, the ISC pathway provides a yet unknown precursor molecule as a sulfur provider for the cytosolic CIA Fe-S cluster assembly pathway [24]. The ISC pathway thus not only governs the proper assembly of mitochondrial Fe-S proteins, but also of cytoplasmic and nuclear ones. Our quantitative proteomics data suggests it is also the case in *T. gondii*, as several putative nuclear Fe-S proteins involved in gene transcription (such as DNA-dependent RNA polymerases) or DNA repair (like DNA endonulease III) were found to be impacted by TgISU1 depletion. The CIA pathway has recently been shown to be important for tachyzoite proliferation [25], and several of the cytoplasmic or nuclear Fe-S cluster-containing proteins are likely essential for parasite viability. It is thus possible that, in spite of their conversion to a stress-resistant form, the long-term viability of TgISU1 parasites could be affected beyond recovery. In vivo experiment in the mouse model may be used to assess this.

The transition from tachyzoite to bradyzoite is known to involve a considerable change in gene expression [65,66], and it takes several days of in vitro differentiation-inducing conditions to obtain mature cysts [99,100]. Quantitative proteomic analysis showed that TgISU1-depleted parasites rapidly display a high expression of bradyzoite-specific surface antigens and GRA markers (S4 Table and Fig 9C). In all the mitochondrial mutants investigated in this study, long term perturbation of the organelle led to the appearance of parasite-containing structures with typical cyst-like morphology (Figs 5, 9 and 10). However, using specific antibodies against early or late bradyzoite markers, the differentiating parasites never appeared to reach a fully mature bradyzoite stage (Figs 9 and 10, S11 Fig). Also, contrarily to the mutants where the ETC may be more directly impacted (TgQCR11 and TgISU1), the mitochondrial translation mutant (TgmS35), while seemingly initiating conversion to bradyzoite, then continued to multiply to some extent and lyse its host cells, likely as tachyzoites (Fig 10A). However, as the mitochondrion encodes three proteins that are subunits of mitochondrial ETC complexes III (cytochrome b) and IV (CoxI and III), this mutant should also be supposedly impacted for ETC function. This apparent discrepancy might then be explained by a looser control of down-regulation of protein expression, or by compensatory mechanisms that may be specifically at play in this particular mutant. It is also possible that more directly targeting the ETC is a much stronger inducer of differentiation, and thus different mitochondrial mutants may behave differently regarding stage conversion. Importantly, one main reason potentially explaining the incomplete differentiation of these mitochondrial mutants is that

they were generated in a laboratory-adapted type I *T. gondii* strain (RH) that typically does not form cysts: these type I tachyzoites may upregulate specific bradyzoite genes and produce bradyzoite-specific proteins or cyst wall components, but they are largely incapable of forming mature bradyzoite cysts [101] (S11 Fig). This calls for further investigations, and in particular generating similar mitochondrion-related mutants in more cystogenic type II parasites, which could be very insightful.

Our quantitative proteomics analysis shows that SUF-impaired parasites also seem to initiate an upregulation of some bradyzoite markers early after TgNFS2 depletion. Yet, these parasites did not display the hallmarks of bradyzoite morphology. They did not progress towards stage conversion and instead they displayed considerable perturbation of the cell division process (Fig 7B), and eventually died. Both the apicoplast and the mitochondrion have established a close metabolic symbiosis with their host cell, so there are likely multiple mechanisms allowing these organelles to communicate their status to the rest of the cell. This raises the question as to why mitochondrion, but not apicoplast, dysfunction can lead to differentiation into bradyzoites. This may be due to differences in the kinetics or the severity of apicoplast-related phenotypes that may not allow stage conversion (which is typically a long process) to happen. Alternatively, there might be differentiation signals specifically associated to the mitochondrion. In fact this organelle is increasingly seen as a signalling platform, able to communicate its fitness through the release of specific metabolites, reactive oxygen species, or by modulating ATP levels [102]. Interestingly, it was shown in other eukaryotes that mitochondrial dysfunctions, such as altered oxidative phosphorylation, significantly impair cellular proliferation, oxygen sensing or specific histone acetylation, yet without diminishing cell viability and instead may lead to quiescent states [103–105]. Consequences of mitochondrial dysfunction include a restricted energy supply and thus constitutes a metabolic challenge that can trigger important cellular adaptations that ultimately determine eukaryotic cell fate and survival. Our results suggest that this is also possibly the case for *T. gondii*. It is for instance quite interesting to see that altering the respiratory activity of the organelle, whether it is by generating specific mutants or by the use of drugs such as atovaquone, seems to lead to a similar differentiation phenotype.

More generally, the environmental or metabolic cues that drive specific gene expression to induce a functional shift leading to conversion into bradyzoites are not fully identified. Moreover, how these stimuli are integrated is also largely unknown. A high-throughput approach has allowed the recent identification of a master transcriptional regulator of stage conversion [106], but how upstream events are converted into cellular signals to mobilize the master regulator is still an important, yet unresolved, question. Translational control [107] may play a role in regulating this factor in the context of the integrated stress response [108]. In fact, an essential part of the eukaryotic cell stress response occurs post-transcriptionally and is achieved by RNA-binding proteins [109]. Interestingly, among the proteins significantly less abundant in the spontaneously-differentiating TgISU1 mutant were many RNA-binding proteins. They include components of stress granules (PolyA-binding protein, PUF1, Alba1 and 2, some of which are linked to stage conversion [110–112]), which are potentially involved in mRNA sequestration from the translational machinery, but also two regulators of the large 60S ribosomal subunit assembly, as well as the gamma subunit of the eukaryotic translation initiation factor (eIF) complex 4 (known to be down-regulated in the bradyzoite stage [113]). Variation in these candidates may have a considerable impact on the translational profile and on the proteostasis of differentiating parasites, and how they may help regulating stage conversion in this context should be investigated further. Understanding the mechanisms that either lead to encystment or death of the parasites is crucial to the development of treatments against toxoplasmosis. This question is key to the pathology, because long term persistence of bradyzoites

and their resistance to current treatments makes them a durable threat for their human hosts. Comparative studies of stress-induced or spontaneously differentiating conditional mutants may bring further insights on how the parasites integrate upstream stresses or dysfunctions into global regulation of stage conversion.

## Materials and methods

### Parasites and cells culture

Tachyzoites of the type I RH TATi RHΔKu80 *T. gondii* cell line [44], as well as derived transgenic parasites generated in this study and the type II Prugniaud strain, were maintained by serial passage in human foreskin fibroblast (HFF, American Type Culture Collection, CRL 1634) cell monolayer grown in Dulbecco's modified Eagle medium (DMEM, Gibco), supplemented with 5% decomplemented fetal bovine serum (FBS), 2-mM L-glutamine and a cocktail of penicillin-streptomycin at 100 μg/ml. The TgQCR11 [63] and TgmS35 [77] conditional mutants cell lines were generously provided by the Sheiner laboratory.

### Bioinformatic analyses

Sequence alignments were performed using the MUltiple Sequence Comparison by Log-Expectation (MUSCLE) algorithm of the Geneious 6.1.8 software suite (http://www.geneious.com). Transit peptide and localization predictions were done using IPSORT (http://ipsort.hgc.jp/), Localizer 1.0.4 (http://localizer.csiro.au/), and Deeploc 1.0 (http://www.cbs.dtu.dk/services/DeepLoc-1.0/) algorithms.

The putative Fe-S proteome was predicted using the MetalPredator webserver (http://metalweb.cerm.unifi.it/tools/metalpredator/) [56]. The whole complement of *T. gondii* annotated proteins was downloaded in FASTA format from the ToxoDB database (https://toxodb.org [37], release 45) and used for analysis in the MetalPredator webserver. Additional manual curation included homology searches for known Fe-S proteins from plants (see appendix A in [114]), and search for homologues in the Uniprot database (https://www.uniprot.org) that were annotated as containing a Fe-S cofactor. For proteomics candidates, annotations were inferred from ToxoDB, KEGG (https://www.genome.jp/kegg/) and the Liverpool Library of Apicomplexan Metabolic Pathways (http://www.llamp.net/ [115]).

N-glycosylation predictions were done with the GlycoEP webserver (http://crdd.osdd.net/raghava/glycoep/index.html). GPI anchor predictions were done with the PredGPI (http://gpcr.biocomp.unibo.it/predgpi/) and GPI-SOM (http://gpi.unibe.ch/) webservers.

Candidate variant proteins identified by quantitative proteomics were mapped on the graphic representation of the high-resolution spatial proteome map of *T. gondii* using the hyperLOPIT dataset [38] (https://proteome.shinyapps.io/toxolopittzex/).

### Heterologous expression in *E. coli*

Constructs for designing recombinant proteins were defined by aligning TgNFS2 and TgISU1 amino acid sequences with their *E. coli* counterparts. For *TgNFS2*, a 1,438 bp fragment corresponding to amino acids 271–699, was amplified by polymerase chain reaction (PCR) from *T. gondii* cDNA using primers ML4201/ML4012 (sequences of the primers used in this study are found in S6 Table). For *TgISU1*, a 393 bp fragment corresponding to amino acids 64–194, was amplified by PCR from *T. gondii* cDNA using primers ML4204/ML4205. The fragments were cloned into the pUC19 (Thermo Fisher Scientific) using the HindIII/BamHI and SphI/BamHI restriction sites, respectively. *E. coli* mutants from the Keio collection (obtained from the The *Coli* Genetic Stock Center at the University of Yale: stain numbers JW1670-1 for *SufS*,

JW2513-1 for *IscU*), were transformed with plasmids for expressing recombinant TgNFS2 and TgISU1 and selected with ampicillin. For growth assays [41], overnight stationary phase cultures were adjusted to the same starting $OD_{600}$ of 0.6 in salt-supplemented M9 minimal media containing 0.4% glucose and varying amounts of the 2,2′-Bipyridyl iron chelator (Sigma-Aldrich). Parental *E. coli* (strain K12) were included as a control. Growth was monitored through $OD_{600}$ measurement after 7, 14 and 24 hours at 37˚C in a shaking incubator.

## Generation of HA-tagged TgNFS2, TgSUFE2 and TgISU1 cell lines

The ligation independent strategy [43] was used for C-terminal hemagglutinin $(HA)_3$-tagging of TgISU1. A fragment corresponding to the 3' end of *TgISU1* was amplified by PCR from genomic DNA, with the Q5 DNA polymerase (New England BioLabs) using primers ML4208/ML4209 and inserted in frame with the sequence coding for a triple HA tag, present in the pLIC-HA₃-chloramphenicol acetyltransferase (CAT) plasmid. The resulting vector was linearized and 40 µg of DNA was transfected into the TATi ΔKu80 cell line to allow integration by single homologous recombination, and transgenic parasites of the TgISU1-HA cell line were selected with chloramphenicol and cloned by serial limiting dilution.

For TgNFS2 and TgSUFE2, a CRISPR-based strategy was used. Using the pLIC-HA₃-CAT plasmid as a template, a PCR was performed with the KOD DNA polymerase (Novagen) to amplify the tag and the resistance gene expression cassette with primers ML3978/ML3979 (*TgNFS2*) and ML4023/ML4162 (*TgSUFE2*), that also carry 30 bp homology with the 3′ end of the corresponding genes. A specific single-guide RNA (sgRNA) was generated to introduce a double-stranded break at the 3′ of the respective loci. Primers used to generate the guides were ML3948/ML3949 (*TgNFS2*) and ML4160/ML4161 (*TgSUFE2*) and the protospacer sequences were introduced in the Cas9-expressing pU6-Universal plasmid (Addgene, ref #52694) [39]. Again, the TATi ΔKu80 cell line was transfected and transgenic parasites of the TgNFS2-HA or TgSUFE2-HA cell lines were selected with chloramphenicol and cloned by serial limiting dilution.

## Generation of TgNFS2 and TgISU1 conditional knock-down and complemented cell lines

The conditional knock-down cell lines for *TgNFS2* and *TgISU1* were generated based on the Tet-Off system using the DHFR-TetO7Sag4 plasmid [116].

For *TgISU1*, a 930 bp 5' region of the gene, starting with the initiation codon, was amplified from genomic DNA by PCR using Q5 polymerase (New England Biolabs) with primers ML4212/ML4213 and cloned into the DHFR-TetO7Sag4 plasmid, downstream of the anhydrotetracycline (ATc)-inducible TetO7Sag4 promoter, yielding the DHFR-TetO7Sag4-TgISU1 plasmid. The plasmid was then linearized and transfected into the TgISU1-HA cell line. Transfected parasites were selected with pyrimethamine and cloned by serial limiting dilution. A similar approach was used to generate an independent cKD TgISU1 cell line, but starting from the TATi ΔKu80 cell line instead of the TgISU1-HA cell line, to allow subsequent tagging of mitochondrial candidates.

For *TgNFS2*, a CRISPR-based strategy was used. Using the DHFR-TetO7Sag4 plasmid as a template, a PCR was performed with the KOD DNA polymerase (Novagen) to amplify the promoter and the resistance gene expression cassette with primers ML4154/ML4155 that also carry 30 bp homology with the 5′ end of the *TgNFS2* gene. A specific sgRNA was generated to introduce a double-stranded break at the 5′ of the *TgNFS2* locus. Primers used to generate the guide were ML4156/ML4157 and the protospacer sequences were introduced in the pU6-Universal plasmid (Addgene ref#52694) [39]. The *TgNFS2*-HA cell line was transfected with the

donor sequence and the Cas9/guide RNA-expressing plasmid and transgenic parasites were selected with pyrimethamine and cloned by serial limiting dilution.

The cKD TgNFS2-HA and cKD TgISU1-HA cell lines were complemented by the addition of an extra copy of the respective genes put under the dependence of a *tubulin* promoter at the *uracil phosphoribosyltransferase* (*UPRT*) locus. *TgNFS2* (2097 bp) and *TgISU1* (657 bp) whole cDNA sequences were amplified by reverse transcription (RT)-PCR with primers ML4576/ML4577 and ML4455/ML4456, respectively. They were then cloned downstream of the *tubulin* promoter sequence of the pUPRT-TUB-Ty vector [44] to yield the pUPRT-TgNFS2 and pUPRT-TgISU1plasmids, respectively. These plasmids were then linearized prior to transfection of the respective mutant cell lines. The recombination efficiency was increased by co-transfecting with the Cas9-expressing pU6-UPRT plasmids generated by integrating *UPRT*-specific protospacer sequences (with primers ML2087/ML2088 for the 3', and primers ML3445/ML3446 for the 5') which were designed to allow a double-strand break at the *UPRT* locus. Transgenic parasites were selected using 5-fluorodeoxyuridine and cloned by serial limiting dilution to yield the cKD TgNFS2-HA comp cKD TgISU1-HA comp cell lines, respectively.

## Generation of HA-tagged TgSDHB and TgApiCox13 cell lines

A CRISPR-based strategy was used. Using the pLIC-HA$_3$-CAT plasmid as a template, a PCR was performed with the KOD DNA polymerase (Novagen) to amplify the tag and the resistance gene expression cassette with primers ML5116/ML5117 (*TgSDHB*) and ML5114/ML5115 (*TgApiCox13*), that also carry 30 bp homology with the 3' end of the corresponding genes. A specific sgRNA was generated to introduce a double-stranded break at the 3' of the respective loci. Primers used to generate the guides were ML4986/ML4987 (*TgSDHB*) and ML4984/ML4985 (*TgApiCox13*) and the protospacer sequences were introduced in the Cas9-expressing pU6-Universal plasmid (Addgene, ref #52694) [39]. The cKD TgISU1 cell line was transfected and parasites were selected with chloramphenicol and cloned by serial limiting dilution.

## Immunoblot analysis

Protein extracts from $10^7$ freshly egressed tachyzoites were prepared in Laemmli sample buffer, separated by SDS-PAGE and transferred onto nitrocellulose membrane using the BioRad Mini-Transblot system according to the manufacturer's instructions. Rat monoclonal antibody (clone 3F10, Roche) was used to detect HA-tagged proteins. Other primary antibodies used were rabbit anti-lipoic acid antibody (ab58724, Abcam), mouse anti-SAG1 [117], rabbit anti-CPN60 [118] and mouse anti-actin [119]. Protein quantification was performed by band densitometry using FIJI (https://imagej.net/software/fiji/).

## Immunofluorescence microscopy

For immunofluorescence assays (IFA), intracellular tachyzoites grown on coverslips containing HFF monolayers, were either fixed for 20 min with 4% (w/v) paraformaldehyde (PFA) in PBS and permeabilized for 10 min with 0.3% Triton X-100 in PBS or fixed for 5 min in cold methanol (for the use of cyst-specific antibodies). Slides/coverslips were subsequently blocked with 0.1% (w/v) BSA in PBS. Primary antibodies used (at 1/1,000, unless specified) to detect subcellular structures were rabbit anti-CPN60 [118], mouse monoclonal anti-F1-ATPase beta subunit (gift of P. Bradley), mouse monoclonal anti-GRA3 [120], rabbit anti-TgHSP28 [121], rabbit anti-GAP45 [122], mouse monoclonal anti-SAG1 [117], anti P18/SAG4 (diluted 1/200, T8 3B1) and anti P21 (diluted 1/200, T8 4G10) [123]. Rat monoclonal anti-HA antibody

(clone 3F10, Roche) was used to detect epitope-tagged proteins. Staining of DNA was performed on fixed cells by incubating them for 5 min in a 1 µg/ml 4,6-diamidino-2-phenylindole (DAPI) solution. All images were acquired at the Montpellier RIO imaging facility from a Zeiss AXIO Imager Z1 epifluorescence microscope driven by the ZEN software v2.3 (Zeiss). Z-stack acquisition and maximal intensity projection was performed to visualize larger structures such as in vitro cysts. Adjustments for brightness and contrast were applied uniformly on the entire image.

## Plaque assay

Confluent monolayers of HFFs were infected with freshly egressed parasites, which were left to grow for 7 days in the absence or presence of ATc (added to a final concentration of 1 µg/ml). They were then fixed with 4% v/v PFA and plaques were revealed by staining with a 0.1% crystal violet solution (V5265, Sigma-Aldrich). For the washout experiments, after 7 days the culture medium was removed from the wells and one gentle wash was performed with Hanks' Balanced Salt Solution (HBSS), taking care not to disturb the cells; then new DMEM medium containing or not ATc was added for another 7-day incubation, before termination of the experiment by PFA fixation and crystal violet staining.

## Egress assay

*T. gondii* tachyzoites were grown for 40 (without ATc) or 120 (with ATc) hours on HFF cells with coverslips in 24-well plates. The infected host cells were incubated for 7 min at 37˚C with DMEM containing 3 µM of calcium ionophore A23187 (C7522, Sigma-Aldrich) prior to fixation with 4% PFA. Immunofluorescence assays were performed as previously described [124]: the parasites and the parasitophorous vacuole membrane were labelled with anti-GAP45 and anti-GRA3, respectively. The proportion of egressed and non-egressed vacuoles was calculated by counting 250 vacuoles in three independent experiments. Data are presented as mean values ± SEM.

## Alkaline stress-induced stage conversion

Tachyzoites were seeded onto HFF monolayers grown on coverslips in 24-well plates. The inoculum was adapted to the cell line used to avoid early lysis of the host cells, with as much as 20,000 parasites per well for the type II Prugniaud strain, and as little as 300 for the type I RH TATi ΔKu80 cell line. 24 hours after infection, in order to induce to induce alkaline stress as previously described [76], the regular culture medium was changed to a pH 8.2 differentiation medium: Minimum Essential Medium (MEM) without $NaHCO_3$ (Gibco), 3% FBS, 50 mM HEPES, pH 8.2. Cultures were maintained at 37˚C in air (to avoid pH variation due to $CO_2$), and the medium was changed every 48 hours for the whole duration of the experiment.

## Semi-quantitative RT-PCR

Total mRNAs of freshly egressed extracellular parasites from the cKD TgNFS2-HA, cKD TgISU1-HA and their respective complemented cell lines, as well as cKD TgISU1 (incubated with or without ATc at 1.5 µg/mL for 3 days) were extracted using Nucleospin RNA II Kit (Macherey-Nagel). The cDNAs were synthesized with 450 ng of total RNA per RT-PCR reaction using High-Capacity cDNA Reverse Transcription Kit (Applied Biosystems). Specific primers for *TgNFS2* (ML4686/ML4687), *TgISU1* (ML4684/ML4685) and, as a control, *Tubulin β* (ML841/ML842) or *actin* (ML843/ML844) were used to amplify specific transcripts with the

GoTaq DNA polymerase (Promega). PCR was performed with 21 cycles of denaturation (30 s, 95°C), annealing (20 s, 55°C), and elongation (30 s, 72°C).

## Mitochondrial membrane potential measurement

Parasites grown for the indicated time with or without ATc were mechanically released from their host cells, purified on a glass wool fiber column, washed and adjusted to $10^7$ parasites/ml in phenol red-free medium, and incubated in with 1.5 µM of the JC-1 dye (5,5',6,6'-tetra-chloro-1,1',3,3'-tetraethylbenzimidazolylcarbocyanine Iodide, T3168, Invitrogen) for 30 min at 37°C, washed phenol red-free medium and analyzed by flow cytometry or microscopy. Flow cytometry analysis was performed on a FACSAria III flow cytometer (Becton Dickinson). An unstained control was used to define gates for analysis. 50,000 events per condition were collected and data were analysed using the FlowJo Software.

## Quantitative label-free mass spectrometry

Parasites of the TATi ΔKu80 and cKD TgISU1-HA cell lines were grown for two days in the presence of ATc; parasites of the cKD TgNFS2-HA were grown for three days in the presence of ATc. Then they were mechanically released from their host cells, purified on a glass wool fiber column, washed in Hanks' Balanced Salt Solution (Gibco). Samples were first normalized on parasite counts, but further adjustment was performed after parasite pellet resuspension in SDS lysis buffer (50 mm Tris-HCl pH8, 10 mm EDTA pH8, 1% SDS) and protein quantification with a bicinchoninic acid assay kit (Abcam). For each condition, 20 µg of total proteins were separated on a 12% SDS-PAGE run for 20 min at 100 V, stained with colloidal blue (Thermo Fisher Scientific), and each lane was cut in three identical fractions. Trypsin digestion and mass spectrometry analysis in the Q Exactive Plus mass spectrometer (Thermo Fisher Scientific) were carried out as described previously [125].

For peptide identification and quantification, the raw files were analyzed with MaxQuant version 1.6.10.43 using default settings. The minimal peptide length was set to 6. Carbamido-methylation of cysteine was selected as a fixed modification and oxidation of methionine, N-terminal-pyroglutamylation of glutamine and glutamate and acetylation (protein N terminus) as variable modifications. Up to two missed cleavages were allowed. The files were searched against the *T. gondii* proteome (March 2020 -https://www.uniprot.org/proteomes/UP000005641-8450entries). Identified proteins were filtered according to the following criteria: at least two different trypsin peptides with at least one unique peptide, an *E* value below 0.01 and a protein E value smaller than 0.01 were required. Using the above criteria, the rate of false peptide sequence assignment and false protein identification were lower than 1%. Peptide ion intensity values derived from MaxQuant were subjected for label-free quantitation. Unique and razor peptides were considered [126]. Statistical analyses were carried out using R package software. ANOVA test with threshold of 0.05 was applied to identify the significant differences in the protein abundance. Hits were retained if they were quantified in at least three of the four replicates in at least one experiment. Additional candidates that consistently showed absence or presence of LFQ values versus the control, and mean LFQ was only considered if peptides were detected in at least three out of the four biological replicates.

## Statistical analysis for phenotypic assays

Unless specified, values are usually expressed as means ± standard error of the mean (SEM). Data were analysed for comparison using unpaired Student's t-test with equal variance (homo-scedastic) for different samples or paired Student's t-test for similar samples before and after

treatment. For comparisons with of ratios of groups of samples with a reference set to 1, analysis of variance (ANOVA) was used.

## Supporting information

**S1 Fig. Alignment of SufS/NFS2 and IscU/ISU1 homologs.** TgNFS2 (A) and TgISU1 (B) homologs were aligned to their counterparts from plant (*Arabidopsis thaliana*) and bacteria (*Escherichia coli*). Key conserved cysteine residue for cysteine desulfurase activity is indicated.
(PDF)

**S2 Fig. Generation of HA-tagged TgNFS2 and TgISU1 cell lines.** A) Schematic representation of the strategy for expressing HA-tagged versions of TgNFS2 (left) and TgISU1 (right) by homologous recombination at the native locus of the corresponding gene of interest. Chloramphenicol was used to select transgenic parasites based on their expression of the Chloramphenicol acetyltransferase (CAT). B) Diagnostic PCR for verifying correct integration of the construct. The amplified fragments correspond to the blue or red arrows in A), and specific primers used were ML3982/ML1476 (TgNFS2) and ML4208/ML1476 (TgISU1).
(PDF)

**S3 Fig. HA-tagging of TgSUFE2 shows it is an apicoplast protein.** A) Sequence alignment of TgSUFE2 (TGGT1_277010) with plant (*A. thaliana*) and bacterial (*E. coli*) homologues. B) Schematic representation of the strategy for expressing an HA-tagged version of TgSUFE2 by double homologous recombination at the native locus. Chloramphenicol was used to select transgenic parasites based on their expression of the Chloramphenicol acetyltransferase (CAT). C) Diagnostic PCR for verifying correct integration of the construct. The amplified fragment corresponds to the red arrows in B), and specific primers used were ML4101/ML1476. D) Detection by immunoblot of C-terminally HA-tagged TgSUFE2 in parasite extracts reveals the presence of both precusor and mature forms of the protein. Anti-actin antibody (TgACT1) was used as a loading control. E) Immunofluorescence assay shows TgSUFE2 co-localizes with apicoplast marker TgCPN60. Scale bar represents 5 μm. DNA was labelled with DAPI. DIC: differential interference contrast.
(PDF)

**S4 Fig. Generation of TgNFS2 and TgISU1 conditional mutants.** A) Schematic representation of the strategy for generating TgNFS2 (top) and TgISU1 (bottom) conditional knockdown cell lines by homologous recombination at the native locus. Pyrimethamine was used to select transgenic parasites based on their expression of Dihydrofolate reductase (DHFR). B) Diagnostic PCR for verifying correct integration of the construct. The amplified fragments confirming 5' and 3' integration correspond to the blue and red arrows displayed in A), respectively, and specific primers used were: ML4158/ML687 (TgNFS2 5' integration), ML1041/ML4159 (TgNFS2 3' integration), ML1774/ML4388 (TgISU1 5' integration), ML1771/ML4387 (TgISU1 3' integration).
(PDF)

**S5 Fig. Generation of TgNFS2 and TgISU1 complemented cell lines.** A) Schematic representation of the strategy for generating TgNFS2 and TgISU1 complemented cell lines by integrating an extra copy of the gene of interest (GOI) by double homologous recombination at the *Uracil Phosphoribosyltransferase* (*UPRT*) locus. Negative selection with 5-fluorodeoxyuridine (FUDR) was used to select transgenic parasites based on their absence of UPRT expression. B) Diagnostic PCR for verifying correct integration of the construct. The amplified fragments confirming integration correspond to the red arrows displayed in A), and specific primers

used were: ML2866/ML4686 (TgNFS2 integration), ML2866/ML4455 (TgISU1 5' integration).
C) Semi-quantitative RT-PCR analysis from cKD TgNFS2-HA, cKD TgISU1-HA and their
respective complemented cell lines grown for three days in the presence or absence of ATc,
using specific primers couples ML4686/ML4687 *(TgNFS2)* and ML4684/ML4685 (*TgISU1*). It
shows complemented cell lines express high levels of the corresponding mRNA. Specific *β-tubulin* primers (ML841/ML842) were used as controls.
(PDF)

**S6 Fig. Quantification of apicoplast loss upon TgNFS2 depletion using streptavidin.** Percentage of cKD TgNFS2-HA parasites-containing vacuoles displaying a loss of apicoplast signal when labeled with streptavidin-Fluorescein Isothiocyanate after culture in the presence or absence of ATc for 120 hours. Data are mean values from $n$ = 3 independent experiments ±SEM. **** $p \leq 0.0001$, Student's $t$-test. Inset: typical streptavidin labeling of the apicoplast (green) in a tachyzoites-containing vacuole, the inner membrane complex was stained with anti-TgIMC3 (red) and the DNA with DAPI (blue); scale bar = 5μm.
(PDF)

**S7 Fig. Quantitative proteomics shows depletion of TgNFS2 does not have a global impact on the apicoplast, but may suggest compensatory response from other cellular pathways in response to specific apicoplast-related lipid synthesis defects.** Classification of variant proteins according to their putative cellular localization (A) and function (B). N/A: not available; ER: endoplasmic reticulum; PM: plasma membrane; VAC: vacuolar compartment; GRA: dense granule protein; SRS: SAG-related sequence. In particular, the increased expression of ER-located lipid metabolism enzymes suggests possible compensation for loss of apicoplast-related lipid synthesis function.
(PDF)

**S8 Fig. TgISU1-depleted parasites show a marked decrease in proteins related to mitochondrial respiration, and a strong increase in bradyzoite-specific dense granules proteins and surface antigens.** Classification of variant proteins according to their putative cellular localization (A) and function (B). N/A: not available; ER: endoplasmic reticulum; PM: plasma membrane; VAC: vacuolar compartment; GRA: dense granule protein; SRS: SAG-related sequence. A large proportion of components of complexes II, III and IV of the mitochondrial respiratory chain, which involve Fe-S proteins, were found to be less abundant. Conversely, the abundance of many bradyzoite-specific dense granule proteins of plasma membrane-located surface antigens increased.
(PDF)

**S9 Fig. Generation of a tag-free cKD TgISU1 cell line.** A) Schematic representation of the strategy for generating the conditional knock-down cell line by homologous recombination at the native locus. Pyrimethamine was used to select transgenic parasites based on their expression of Dihydrofolate reductase (DHFR). B) Diagnostic PCR for verifying correct integration of the construct. The amplified fragments confirming 5' and 3' integration correspond to the blue and red arrows displayed in A), respectively, and specific primers used were ML1774/ML4388 (5' integration), and ML1771/ML4387 (3' integration). C) Semi-quantitative RT-PCR analysis of the cKD TgISU1 cell line grown for up to three days in the presence or absence of ATc, using specific primers couple ML4684/ML4685, showing effcient down-regulation of *TgISU1* expression. Specific *actin* primers (ML843/ML844) were used as controls. D) Plaque assays were carried out by infecting HFF monolayers with the newly generated cKD TgISU1 cell line or the original cKD TgISU1-HA mutant cell line as a control. They were grown for 7 days ± ATc. Measurements of lysis plaque areas are shown on the right

and confirm a significant defect in the lytic cycle in the two mutant cell lines upon ATc addition. Values are means of n = 3 experiments ± SEM. **** denotes $p \leq 0.0001$, ANOVA. Scale bar = 1mm.
(PDF)

**S10 Fig. Tagging of mitochondrial candidates in the cKD TgISU1 background.** A) Schematic representation of the strategy for expressing HA-tagged versions of TgSDHB and TgApiCox13 by homologous recombination at the native locus of the corresponding gene of interest. Chloramphenicol was used to select transgenic parasites based on their expression of the Chloramphenicol acetyltransferase (CAT). B) Diagnostic PCR for verifying correct integration of the construct. The amplified fragment corresponds to the red arrows in A), and specific primers used were ML1476/ML5013 (TgSDHB), ML1476/ML5006 (TgApiCox13).
(PDF)

**S11 Fig. Comparison of stage conversion for type I and II strains.** A) Stage conversion was induced by alkaline pH stress on type I parasites of the RH Tati Δ80 parental cell line or the cystogenic type II Prugniaud strain for up to 14 days. Fixed samples were co-stained with cyst wall marker DBL, together with tachyzoite marker SAG1, or intermediate (P18/SAG4) or late (P21) bradyzoite markers. Scale bar represents 10 µm. DNA was labelled with DAPI. B) The percentage of DBL-positive cysts containing P18 or P21 staining was evaluated on samples after 14 days of ATc treatment (for TgISU1, TgQCR11 and TgmS35 conditional mutants) or pH stress (for RH TATi ΔKu80 and Prugniaud parasites). Values are mean ±SEM from $n = 3$ independent experiments. C) Measurement of the cyst area size after growing the cell lines for 7 and 14 days in cyst-inducing conditions, then labelling the cyst wall with DBL and measuring the surface of at least 25 cysts per condition. Values are mean ±SD from three independent biological replicates. $^{*}$ $p \leq 0.05$, $^{**}$ $p \leq 0.01$, Student's $t$-test, when comparing values after 14 days between the type II Prugniaud strain and the type I cell lines.
(PDF)

**S1 Table. Predicted Toxoplasma homologues of the iron-sulfur cluster synthesis machinery.** Homology searches were conducted in ToxoBD.org using *Arabidopsis thaliana* proteins as a query. Putative subcellular localization was obtained from the hyperLOPIT data available on ToxoDB.org, or by manual annotation. CRISPR fitness score data was obtained from ToxoDB.org.
(XLSX)

**S2 Table. Predicted Toxoplasma iron-sulfur proteome.** The Toxoplasma predicted whole proteome was obtained from the ToxoDB.org database and searched for putative iron-sulfur-containing proteins with the MetalPredator web server (http://metalweb.cerm.unifi.it/tools/metalpredator/). Putative subcellular localization was obtained from the hyperLOPIT data available on ToxoDB.org, or by manual annotation. CRISPR fitness score data was obtained from ToxoDB.org.
(XLSX)

**S3 Table. Proteins with lower or higher expression upon depletion of TgNFS2 as found by label-free quantitative proteomics.** For each protein candidate (with www.ToxoDB.org and www.Uniprot.org identifier), $\log_2$ of the different ratio were calculated between the mean MaxQuant LFQ values ('moyLFQ') found for the TgISU1 ('Mito') and TgNFS2 ('Apicoplast') mutants, and the TATi ΔKu80 control ('CTRL'). $-\log_{10}$(pvalue) is also provided. Putative subcellular localization was obtained from the hyperLOPIT data available on ToxoDB.org, or by

manual annotation. CRISPR fitness score and transcriptomic data for tachyzoites (Tz) and bradyzoites (Bz) were obtained from ToxoDB.org.
(XLSX)

**S4 Table. Proteins with lower or higher expression upon depletion of TgISU1 as found by label-free quantitative proteomics.** See legend of S3 Table. Candidates from the Fe-S proteome (S2 Table) that were found to have a lower expression upon TgISU1 depletion are highlighted in red.
(XLSX)

**S5 Table. Common proteins with lower or higher expression upon depletion of TgNFS2 or TgISU1, as found by label-free quantitative proteomics.** See legend of S3 Table.
(XLSX)

**S6 Table. Oligonucleotides used in this study.**
(XLSX)

## Acknowledgments

We are grateful to L. Sheiner, P. Bradley, B. Striepen, V. Carruthers, S. Lourido, S. Angel and D. Soldati-Favre for providing cell lines, antibodies and plasmids. We thank the developers and the managers of the VeupathDB.org/ToxoDB.org databases, as well as scientists who contributed datasets. We also thank the MRI imaging facility for providing access to their microscopes and flow cytometers, and the Mass Spectrometry Proteomics Platform (MSPP) of the BPMP laboratory. Thanks to F. Vignols for insights into the biochemistry of Fe-S proteins.

## Author Contributions

**Conceptualization:** Sébastien Besteiro.

**Data curation:** Sébastien Besteiro.

**Formal analysis:** Sarah Pamukcu, Aude Cerutti, Sonia Hem, Valérie Rofidal, Sébastien Besteiro.

**Funding acquisition:** Sébastien Besteiro.

**Investigation:** Sarah Pamukcu, Aude Cerutti, Yann Bordat, Valérie Rofidal, Sébastien Besteiro.

**Methodology:** Sonia Hem.

**Supervision:** Sébastien Besteiro.

**Writing – original draft:** Sébastien Besteiro.

**Writing – review & editing:** Sébastien Besteiro.

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
