## [Decision Letter · Decision Letter 0]

9 Oct 2021

Dear Dr. Besteiro,

Thank you very much for submitting your manuscript "Differential contribution of two organelles of endosymbiotic origin to iron-sulfur cluster synthesis and overall fitness in Toxoplasma" for consideration at PLOS Pathogens.

Your manuscript was re-reviewed by  two independent reviewers who assessed the first iteration of this work . In light of the reviews (below this email), we would like to invite the resubmission of a revised version that takes into account the reviewer #2 comments.

Importantly, the novelty in the study lies primarily in the observation that knockdown of the mitochondrial Fe-S cluster synthesis pathway induces an apparent switch to bradyzoites. According to this reviewer the  characterisation of this differentiation phenotype is still somewhat preliminary, and warrants a more in-depth analysis.

Please prepare and submit your revised manuscript within 30  tdays. If you anticipate any delay, please let us know the expected resubmission date by replying to this email.

Sincerely,

Dominique Soldati-Favre

Section Editor

PLOS Pathogens

Dominique Soldati-Favre

Section Editor

PLOS Pathogens

Kasturi Haldar

Editor-in-Chief

PLOS Pathogens

orcid.org/0000-0001-5065-158X

Michael Malim

Editor-in-Chief

PLOS Pathogens

orcid.org/0000-0002-7699-2064

Reviewer Comments (if any, and for reference):

Reviewer's Responses to Questions

**Part I - Summary**

Reviewer #1: The authors have made substantial changes to the manuscript with new data and interpretation. My comments have been appropriately addressed.

Reviewer #2: The authors have undertaken additional experiments to address several of the major comments that I raised the original manuscript. Specifically, they:

- include a wild type control strain in the bacterial complementation experiments that indicates that complementation of the bacterial mutants with TgNFS2 and TgISU1 restores bacterial proliferation to near wild type levels.

- undertake a longer “wash-out” period and include appropriate controls (i.e. continuous presence in ATc) in the experiments presented in Figure 4E, which provides greater confidence that IscU knockdown parasites remain viable. Additionally, they include new data that demonstrate a similar retention of viability (and apparent differentiation into bradyzoites) in other mitochondrial mutants.

- undertake validation of the downregulation of some candidate mitochondrial Fe-S proteins identified in the global proteomic analyses, lending confidence to the global proteomics approach.

They address most of the other criticisms of the original manuscript to my satisfaction. In my opinion, aspects of the characterization of bradyzoite differentiation upon TgIscU knockdown are still somewhat preliminary. Given that this is one of the most novel findings of the manuscript, some additional experimentation and data quantification would provide a clearer picture of what is occurring upon knockdown of TgIscU (as suggested in my major comments). Specifically, it would be valuable to undertake additional experiments to determine whether the “partial differentiation” observed upon TgIscU knockdown mimics differentiation that would occur upon standard methods of bradyzoite induction (e.g. NO treatment) in the parasite strain used in these experiments, or whether these partial effects are specific to mitochondrial impairment.

**Part II – Major Issues: Key Experiments Required for Acceptance**

Reviewer #1: None. The revision has addressed the points raised by me.

Reviewer #2: 1. Figure 9D and Figure 10D. The experiments here present single images of P18-positive and SAG1-depleted vacuoles. Given the uncertainties in the data about whether the slower growing vacuoles are ‘real’ tissue cysts, these would benefit from quantification (i.e. what percentage of parasite vacuoles are P18-positive following TgISU1 knockdown? Does this mimic the observations of the DBL-positive vacuoles in Fig 9B/10C?). Alternatively, the authors could consider undertaking western blotting to determine whether the various marker proteins are up/down-regulated upon gene knockdown.

2. Figure 9D and Figure 10D. Given that the authors are working with RH/TATi-background parasites that typically don’t readily differentiate to bradyzoites, they should also induce bradyzoite differentiation using a more standard approach (e.g. NO treatment as in Figure 9B) to determine whether markers like P18 and P21 ‘turn on’ in their parasite strain. This would test whether the observation that “stage conversion of these parasites progresses beyond the appearance of early cyst wall markers … but it seems incomplete” (Lines 441-443) is specific to the TgISU1/TgQCR11 mutants, or because the parental strain doesn’t readily undergo switching to bradyzoites.

**Part III – Minor Issues: Editorial and Data Presentation Modifications**

Reviewer #1: Line 108-109: ""... investigations in apicomplexan parasites have been so far almost

exclusively focused on the apicoplast-located SUF pathway [26–30] and mostly in Plasmodium

species." There's a recent publication which includes structure-function analyses of the P. falciparum mitochondrial ISC pathway (Sadik et al. 2021, Molecular Microbiology - https://doi.org/10.1111/mmi.14735). This should be cited.

Reviewer #2: 1. Line 35-36. “our data highlighted distinct changes at the metabolic level.” The study does not undertake any metabolomics, so this statement isn’t supported by the data. It would be truer to state that changes occurred in the abundance of proteins from various metabolic pathways, suggesting changes at the metabolic level. Similarly, later in this paragraph: “the mitochondrial pathway is clearly crucial for maintaining their respiratory capacity.” I don’t disagree that this is a likely possibility, but the study measures respiratory capacity indirectly (by measuring membrane potential). Some caution is warranted here: “the mitochondrial pathway likely supports the maintenance of respiratory capacity” or similar.

2. Figure 1B. What is the statistical test comparing? IscU/SufS mutants vs the strains complemented with the Toxoplasma homologues? Just the ones cultured in the absence of the chelator? Please clarify this.

3. Line 242-243. “The growth kinetics we observed for this mutant are consistent with the “delayed death” effect observed in apicoplast-defective parasites”. It isn’t clear what “growth kinetics” are being referred to here? Is it that growth of the NFS2 mutant was “slowed considerably” following the second invasion cycle (Line 236)? This isn’t quantified in any way. e.g. Figure 4C is presumably measuring growth in the second cycle, but there is no measure of the first cycle from what I can tell. Please clarify.

4. Line 424-425. “This clearly confirmed a strong increase in the expression of bradyzoite-specific SRS in the TgISU1 mutant”. This makes it seem like bradyzoite-specific SRS proteins are upregulated across the board, whereas this is only true of one (SRS44, which is also upregulated in the NFS2 mutant) or maybe two (SRS13). This suggests that the “differentiation” that the authors are observing is different to typical bradyzoite differentiation. These experiments should therefore be described with more precision (i.e. noted that some but not most bradyzoite-specific SRSs are upregulated upon ISU1 knockdown).

5. Line 441-443. “stage conversion of these parasites progresses beyond the appearance of early cyst wall markers, but not only it does so with slow kinetics, but it seems incomplete.” The meaning here isn’t entirely clear. Perhaps “… but does so with slow kinetics and seems incomplete.” Also need to specify what is meant by “slow kinetics” - slower than what? As noted in the major comments, using a parasite strain that has been induced to form bradyzoites using standard approaches would allow for a more relevant comparison.

6. Lines 532-534. “the three putative rhoptry localized candidates significantly less expressed in the TgNFS2 mutant (Table S3) are predicted to be GPI-anchored and/or glycosylated.”. Based on what evidence? Please provide relevant references that these are GPI anchored/glycosylated.

7. Line 542. “long term depletion of TgISU1” – TgNFS2?

8. Line 580-581. “perhaps because of their metabolic flexibility, we have shown that mutant parasites seem to retain the ability to survive as tachyzoites, or initiate a switch to bradyzoites.” The first point (survival as tachyzoites) reproduces previous findings in the TgQCR11 mutant parasites by Hayward et al (ref 61).

9. Line 602. “the ETC mutants (TgQCR11 and TgISU1)”. ISU1 is not an ETC mutant.

10. Line 603-603. “largely continued to thrive”. “Thrive” would imply that the parasites are growing well, which I don’t think the authors mean to imply.

PLOS authors have the option to publish the peer review history of their article (what does this mean?). If published, this will include your full peer review and any attached files.

Reviewer #1: No

Reviewer #2: No

Figure Files:

Data Requirements:

Reproducibility:

References:

---

## [Editor Report · Decision Letter 1]

5 Nov 2021

Dear Dr. Besteiro,

We are pleased to inform you that your manuscript 'Differential contribution of two organelles of endosymbiotic origin to iron-sulfur cluster synthesis and overall fitness in Toxoplasma' has been provisionally accepted for publication in PLOS Pathogens.

Best regards,

Dominique Soldati-Favre

Associated Editor

PLOS Pathogens

Dominique Soldati-Favre

Section Editor

PLOS Pathogens

Kasturi Haldar

Editor-in-Chief

PLOS Pathogens

orcid.org/0000-0001-5065-158X

Michael Malim

Editor-in-Chief

PLOS Pathogens

orcid.org/0000-0002-7699-2064
---

## [Editor Report · Acceptance letter]

15 Nov 2021

Dear Dr. Besteiro,

We are delighted to inform you that your manuscript, "Differential contribution of two organelles of endosymbiotic origin to iron-sulfur cluster synthesis and overall fitness in Toxoplasma," has been formally accepted for publication in PLOS Pathogens.

Best regards,

Kasturi Haldar

Editor-in-Chief

PLOS Pathogens

orcid.org/0000-0001-5065-158X

Michael Malim

Editor-in-Chief

PLOS Pathogens

orcid.org/0000-0002-7699-2064